# Uncertainty Quantification for LLM-Based Survey Simulations

Chengpiao Huang [1]   Yuhang Wu [2]   Kaizheng Wang [1 3]

## Abstract

We investigate the use of large language models (LLMs) to simulate human responses to survey questions, and perform uncertainty quantification to gain reliable insights. Our approach converts imperfect, LLM-simulated responses into confidence sets for population parameters of human responses, addressing the distribution shift between the simulated and real populations. A key innovation lies in determining the optimal number of simulated responses: too many produce overly narrow confidence sets with poor coverage, while too few yield excessively loose estimates. To resolve this, our method adaptively selects the simulation sample size, ensuring valid average-case coverage guarantees. It is broadly applicable to any LLM, irrespective of its fidelity, and any procedure for constructing confidence sets. Additionally, the selected sample size quantifies the degree of misalignment between the LLM and the target human population. We illustrate our method on real datasets and LLMs.

## 1. Introduction

Large language models (LLMs) have demonstrated remarkable capabilities in mimicking human behaviors. Recent studies have leveraged LLMs to simulate human responses in various domains, including economic and social science experiments (Aher et al., 2023; Horton, 2023; Chen et al., 2023; Bisbee et al., 2024; Huang et al., 2024; Yang et al., 2024; Ziems et al., 2024), market research (Brand et al., 2023; Gui & Toubia, 2023; Goli & Singh, 2024; Wang et al., 2024), education (Zelikman et al., 2023; Lu & Wang, 2024), and so on. The typical simulation procedure consists in prompting an LLM with a real or fictional persona as well

as a survey question, and collecting the LLM's responses. Compared to traditional survey methods that recruit and query real people, LLM simulations offer significant advantages in terms of time and cost efficiency, enabling the generation of large-scale synthetic responses with minimal effort.

However, a growing body of evidence suggests that LLMs are not perfectly aligned with the human population, and in some cases, the misalignment can be substantial (Aher et al., 2023; Santurkar et al., 2023). This raises critical concerns about the reliability of insights derived from LLM-generated data. It remains a challenge how to properly simulate human responses using LLMs and how to account for their imperfections when using the simulated samples to make inference about the true human population.

We propose to address this challenge through the lens of *uncertainty quantification*. Specifically, we seek to construct confidence sets for population statistics of human responses based on LLM-generated data. A central question in this process is:

*How many synthetic samples should be generated?*

On one hand, generating too many samples risks overfitting the synthetic distribution, which may deviate from the real human population. On the other hand, generating too few samples yields overly large and uninformative confidence sets. The optimal sample size depends on the discrepancy between the synthetic and real populations — a quantity that is unknown in practice. This necessitates a data-driven approach to determine the appropriate number of simulated responses.

**Main contributions.**   In this paper, we develop a general framework to address these challenges. Our key contributions are as follows:

- (Formulation) We provide a rigorous mathematical framework for uncertainty quantification in LLM-based survey simulations.

- (Methodology) We propose a flexible methodology that transforms simulated responses into valid confidence sets for population parameters of human responses. Our approach adaptively selects the simulation sample

[1]Department of Industrial Engineering and Operations Research, Columbia University [2]Decision, Risk, and Operations Division, Columbia Business School [3]Data Science Institute, Columbia University. Correspondence to: Kaizheng Wang <kaizheng.wang@columbia.edu>.

*Proceedings of the 42nd International Conference on Machine Learning*, Vancouver, Canada. PMLR 267, 2025. Copyright 2025 by the author(s).

size based on the observed misalignment between the LLM and human populations. It is applicable to any LLM, regardless of its fidelity, and can be combined with any method for confidence set construction.

**Related works.** Our work relates to research on assessing the fidelity of LLM simulations and measuring their alignment with real human populations. Prior studies have explored similarity metrics between synthetic and human distributions (Santurkar et al., 2023; He-Yueya et al., 2024; Dominguez-Olmedo et al., 2024; Durmus et al., 2024; Calderon et al., 2025) and Turing-type tests (Argyle et al., 2023; Mei et al., 2024) to evaluate LLM reliability. While these approaches provide valuable insights into LLM misalignment, they do not offer methods for leveraging imperfect LLM simulations to draw reliable conclusions about human populations. In contrast, our work provides a principled approach for constructing confidence sets that account for the inherent discrepancies between LLM-generated and human responses.

Additionally, our work connects to the line of work on model-free statistical inference, including conformal inference (Vovk et al., 2005; Shafer & Vovk, 2008; Bates et al., 2021; Angelopoulos et al., 2024; Kim et al., 2024) and prediction-powered inference (Angelopoulos et al., 2023). At a high level, these methods use labeled data from the true distribution to calibrate imperfect point predictions from an arbitrary black-box model and then construct valid set estimates. Our approach follows a similar spirit. The "features" and "labels" in our setting correspond to the survey questions and their population statistics of human responses, respectively. However, our labels are not directly observable. As a result, the labeled calibration data needed for these methods is not available. Moreover, for every simulation sample size $k$, one can produce a point prediction of the label using $k$ synthetic responses generated by the LLM. As the optimal sample size is not known a priori, there are infinitely many candidate point predictions to choose from. This makes it difficult to apply existing statistical inference methods.

**Outline.** The rest of the paper is organized as follows. Section 2 studies binary response simulation as a warm-up example. Section 3 presents the general problem setup and methodology. Section 4 illustrates our proposed method on real datasets. Section 5 concludes the paper.

**Notation.** We use $\mathbb{Z}_+$ to denote the set of positive integers. For $n \in \mathbb{Z}_+$, define $[n] = \{1, 2, ..., n\}$. For $a, b \in \mathbb{R}$, define $a \wedge b = \min\{a, b\}$ and $a \vee b = \max\{a, b\}$. For non-negative sequences $\{a_n\}_{n=1}^{\infty}$ and $\{b_n\}_{n=1}^{\infty}$, we write $a_n = O(b_n)$ if there exists $C > 0$ such that for all $n$, it holds that $a_n \leq Cb_n$. We write $a_n = \Omega(b_n)$ if $b_n = O(a_n)$.

We write $a_n = \Theta(b_n)$ if $a_n = O(b_n)$ and $a_n = \Omega(b_n)$. The notation $\mathrm{Bernoulli}(p)$ denotes the Bernoulli distribution with mean $p$. The notation $N(\mu, \sigma^2)$ denotes the normal distribution with mean $\mu$ and variance $\sigma^2$.

## 2. Warm-up: Binary Response Simulation

To motivate our problem and methodology, we will start with a simple setting where an LLM simulates binary responses to a survey question. In Section 3, we will present the general problem setup and the general methodology.

### 2.1. Motivating Example: Educational Test

Suppose a school wants to estimate the proportion $\mu \in [0, 1]$ of students that can answer a newly designed test question correctly. It will not only provide insights into student progress but also evaluate the question's effectiveness in differentiating among students with varying levels of understanding. Such information can guide the school in tailoring teaching strategies to better address student needs.

The most direct approach is to give the test to $n$ students and collect their results $y_1, ..., y_n \in \{0, 1\}$, where $y_i$ indicates whether student $i$ answers the question correctly. A point estimate for $\mu$ is the sample mean $\bar{y} = \frac{1}{n} \sum_{i=1}^{n} y_i$. Given $\alpha \in (0, 1)$, we can construct a confidence interval for $\mu$:

$$\left[ \bar{y} - \frac{s}{\sqrt{n}} \Phi^{-1}\left(1 - \frac{\alpha}{2}\right), \ \bar{y} + \frac{s}{\sqrt{n}} \Phi^{-1}\left(1 - \frac{\alpha}{2}\right) \right], \quad (1)$$

where $s = \sqrt{\bar{y}(1 - \bar{y})}$ is the sample standard deviation, and $\Phi$ is the cumulative distribution function (CDF) of $N(0, 1)$. By the Central Limit Theorem (CLT), this interval has asymptotic coverage probability $1 - \alpha$ as $n \to \infty$. As a different approach, one can also use Hoeffding's concentration inequality (e.g., Theorem 2.8 in (Boucheron et al., 2013)) to construct a finite-sample confidence interval

$$\left[ \bar{y} - \sqrt{\frac{\log(2/\alpha)}{2n}}, \ \bar{y} + \sqrt{\frac{\log(2/\alpha)}{2n}} \right], \quad (2)$$

which has at least $(1 - \alpha)$ coverage probability for every $n \in \mathbb{Z}_+$. For simplicity, we will stick to (1) in this section.

Alternatively, the school may use an LLM to simulate students' responses to the question. Compared with directly testing on real students, this approach is more time-efficient and cost-saving. If we prompt the LLM $k$ times with random student profiles, then it generates $k$ synthetic responses, which leads to synthetic outcomes $y_1^{\mathsf{syn}}, ..., y_k^{\mathsf{syn}} \in \{0, 1\}$. We may also compute the sample mean $\bar{y}_k^{\mathsf{syn}} = \frac{1}{k} \sum_{i=1}^{k} y_i^{\mathsf{syn}}$

and the CLT-based confidence interval

$$\mathcal{I}^{\mathsf{syn}}(k) = \left[ \bar{y}_k^{\mathsf{syn}} - \frac{c \cdot s_k^{\mathsf{syn}}}{\sqrt{k}} \Phi^{-1}\left(1 - \frac{\alpha}{2}\right), \right.$$
$$\left. \bar{y}_k^{\mathsf{syn}} + \frac{c \cdot s_k^{\mathsf{syn}}}{\sqrt{k}} \Phi^{-1}\left(1 - \frac{\alpha}{2}\right) \right], \quad (3)$$

where $s_k^{\mathsf{syn}} = \sqrt{\bar{y}_k^{\mathsf{syn}}(1 - \bar{y}_k^{\mathsf{syn}})}$, and $c > 1$ is a scaling parameter. Such a dilation by $c$ is necessary; without it, whenever the LLM-generated data deviates from the student population (even by the slightest amount), the interval $\mathcal{I}^{\mathsf{syn}}(k)$ may never achieve $(1 - \alpha)$ coverage regardless of $k$. We give an example in Appendix B.1.

Due to the misalignment between the LLM and students, the distribution of the synthetic data $\{y_i^{\mathsf{syn}}\}_{i=1}^k$ may be very different from the true response distribution. In this case, the sample mean $\bar{y}^{\mathsf{syn}}$ can be a poor estimate of $\mu$, and $\mathcal{I}^{\mathsf{syn}}(k)$ is generally not a valid confidence interval for $\mu$. In particular, as $k \to \infty$, the interval concentrates tightly around the synthetic mean $\mathbb{E}[y_1^{\mathsf{syn}}]$ and fails to cover the true mean $\mu$. On the other hand, when $k$ is small, the interval becomes too wide to be informative, even though it may cover $\mu$ with high probability. We provide an illustration in Figure 1.

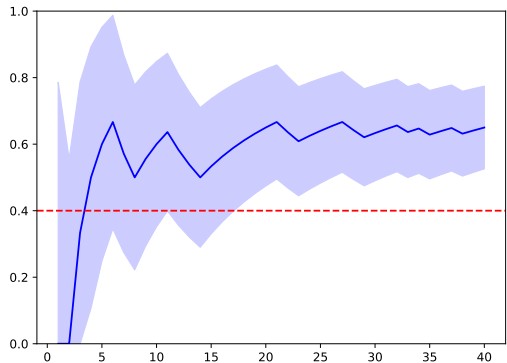

*Figure 1.* The coverage-width trade-off for the simulation sample size $k$. The true distribution is $\mathrm{Bernoulli}(0.4)$ and the synthetic distribution is $\mathrm{Bernoulli}(0.6)$. The horizontal axis is the simulation sample size $k$. The red dotted horizontal line plots the true mean $\mu = 0.4$. The blue curve plots the sample mean $\bar{y}_k^{\mathsf{syn}}$ of the synthetic data, and the blue shaded region visualizes the confidence interval $\mathcal{I}^{\mathsf{syn}}(k)$, for $k \in [40]$. When $k$ is too small (say $k \leq 6$), the interval $\mathcal{I}^{\mathsf{syn}}(k)$ is too wide. When $k$ is too large (say $k \geq 18$), $\mathcal{I}^{\mathsf{syn}}(k)$ fails to cover $\mu$.

**Main insights.** Our goal in this work is to develop a principled approach for choosing a good simulation sample size $\widehat{k}$, so that $\mathcal{I}^{\mathsf{syn}}(\widehat{k})$ is a valid confidence interval for $\mu$ while having a small width. Solving this problem has the following important implications.

1. The choice of $\widehat{k}$ offers valuable information for future simulation tasks on the appropriate number of synthetic samples to generate, so as to produce reliable confidence intervals. It also helps avoid generating excessive samples and improves computational efficiency.

2. The width of $\mathcal{I}^{\mathsf{syn}}(\widehat{k})$ provides an assessment of the alignment between the LLM and the human population. A wide confidence interval indicates high uncertainty of its estimate of the true $\mu$, and thus a large gap between the synthetic data distribution and the true population.

3. The sample size $\widehat{k}$ reflects the size of the target population that the LLM can represent. We make an analogy using the classical theory of parametric bootstrap. Suppose a model is trained via maximum likelihood estimation over $k$ i.i.d. human samples. When performing parametric bootstrap for uncertainty quantification, the bootstrap sample size is usually set to be the training sample size $k$. Thus, our simulation sample size $\widehat{k}$ reveals the LLM as "being made up of" $\widehat{k}$ people from the population. We provide a visualization in Figure 4 of Appendix A. The larger $\widehat{k}$ is, the more diversity that the LLM appears to capture. In contrast, a small $\widehat{k}$ could imply the peculiarity of the LLM compared to the major population.

*Remark* 2.1 (Comparison with existing works). Existing works typically measure LLM misalignment using integral probability metrics and $f$-divergences (Santurkar et al., 2023; Dominguez-Olmedo et al., 2024; Durmus et al., 2024), which do not carry operational meanings themselves and can be hard to interpret. In contrast, our simulation sample size $\widehat{k}$ provides actionable guidance and is easy to understand.

### 2.2. Methodology for Selecting Simulation Sample Size

We now introduce our method for choosing a good simulation sample size $\widehat{k}$. It makes use of similar test questions for which real students' results are available. If such data is available, then we can compare LLM simulations with real students' results on these questions, and use it to guide the choice of $k$.

Specifically, we assume access to $m$ test questions similar to the question of interest. For example, they can come from previous tests or a question bank. For $j \in [m]$, the $j$-th test question has been tested on $n_j$ real students, with test results $\mathcal{D}_j = \{y_{j,i}\}_{i=1}^{n_j}$. We also simulate LLM responses $\mathcal{D}_j^{\mathsf{syn}} = \{y_{j,i}^{\mathsf{syn}}\}_{i=1}^K$ to the $j$-th test question, and $\mathcal{D}^{\mathsf{syn}} = \{y_i^{\mathsf{syn}}\}_{i=1}^K$ to the test question of interest. Here $K \in \mathbb{Z}_+$ is the simulation budget.

For each question $j \in [m]$, we form confidence intervals

similar to (3) using the synthetic data $\mathcal{D}_j^{\mathsf{syn}}$, aiming to cover the true proportion $\mu_j$ of students that can answer the $j$-th question correctly:

$$\mathcal{I}_j^{\mathsf{syn}}(k) = \left[ \bar{y}_{j,k}^{\mathsf{syn}} - \frac{c \cdot s_{j,k}^{\mathsf{syn}}}{\sqrt{k}} \Phi^{-1}\left(1 - \frac{\alpha}{2}\right), \right.$$
$$\left. \bar{y}_{j,k}^{\mathsf{syn}} + \frac{c \cdot s_{j,k}^{\mathsf{syn}}}{\sqrt{k}} \Phi^{-1}\left(1 - \frac{\alpha}{2}\right) \right], \quad (4)$$

where $\bar{y}_{j,k}^{\mathsf{syn}} = \frac{1}{k} \sum_{i=1}^{k} y_{j,i}^{\mathsf{syn}}$ is the sample mean of the first $k$ samples in $\mathcal{D}_j^{\mathsf{syn}}$, and $s_{j,k}^{\mathsf{syn}} = \sqrt{\bar{y}_{j,k}^{\mathsf{syn}}(1 - \bar{y}_{j,k}^{\mathsf{syn}})}$ is the estimated standard deviation. We also set the convention $\mathcal{I}^{\mathsf{syn}}(0) = \mathcal{I}_j^{\mathsf{syn}}(0) = [0,1]$, as nothing can be said about the true parameter without data. We will pick $\hat{k} \in \{0, 1..., K\}$ such that $\mathcal{I}_j^{\mathsf{syn}}(\hat{k})$ covers $\mu_j$ with high probability. We expect this choice of $\hat{k}$ to be also good for $\mathcal{I}^{\mathsf{syn}}(k)$, as the test questions are similar.

Ideally, we would like to pick $k$ such that $(1 - \alpha)$-coverage is achieved empirically over the $m$ test questions:

$$\frac{1}{m} \sum_{j=1}^{m} \mathbf{1}\{\mu_j \notin \mathcal{I}_j^{\mathsf{syn}}(k)\} \leq \alpha. \quad (5)$$

As the true $\{\mu_j\}_{j=1}^m$ are not available, we use the real data $\{\mathcal{D}_j\}_{j=1}^m$ to compute the sample means $\bar{y}_j = \frac{1}{n_j} \sum_{i=1}^{n_j} y_{j,i}$ as proxies for $\mu_j$. The empirical miscoverage can be approximated by

$$G(k) = \frac{1}{m} \sum_{j=1}^{m} \mathbf{1}\{\bar{y}_j \notin \mathcal{I}_j^{\mathsf{syn}}(k)\}. \quad (6)$$

Our criterion for selecting $k$ is given by

$$\hat{k} = \max\{0 \leq k \leq K : G(i) \leq \alpha/2 \ \forall i \leq k\}. \quad (7)$$

Note that $\hat{k}$ is well-defined because $G(0) = 0$.

The choice of the threshold $\alpha/2$ in (7) can be explained as follows. By CLT, when $n_j$ is large, $\mathbb{P}(\mu_j \geq \bar{y}_j) \approx 1/2$. Suppose $\mu_j \notin \mathcal{I}_j^{\mathsf{syn}}(k)$, then $\mu_j$ is either on the left or the right of $\mathcal{I}_j^{\mathsf{syn}}(k)$. In the former case, $\bar{y}_j$ is on the left of $\mu_j$ with probability around $1/2$, which implies $\bar{y}_j \notin \mathcal{I}_j^{\mathsf{syn}}(k)$. Similarly, in the latter case, $\bar{y}_j$ is on the right of $\mu_j$ with probability around $1/2$, and then $\bar{y}_j \notin \mathcal{I}_j^{\mathsf{syn}}(k)$. Roughly speaking, the frequency of having $\bar{y}_j \notin \mathcal{I}_j^{\mathsf{syn}}(k)$ is at least half of the frequency of having $\mu_j \notin \mathcal{I}_j^{\mathsf{syn}}(k)$. In other words, the lower bound

$$G(k) \geq \frac{1}{2} \cdot \frac{1}{m} \sum_{j=1}^{m} \mathbf{1}\{\mu_j \notin \mathcal{I}_j^{\mathsf{syn}}(k)\} \quad (8)$$

approximately holds. Substituting (8) into (5) yields the threshold $\alpha/2$ for choosing $\hat{k}$.

## 2.3. Theoretical Analysis

In this section, we present a theoretical analysis of our proposed method. To do so, we first describe the setup in Section 2.1 and Section 2.2 in mathematical terms.

The student population can be represented by a distribution $\mathcal{P}$ over a space $\mathcal{Z}$ of possible *student profiles*, say, vectors of background information, classes taken, grades, etc. To simulate student responses from the LLM, synthetic student profiles are generated from a synthetic student population $\mathcal{P}^{\mathsf{syn}}$ over $\mathcal{Z}$, and then fed to the LLM.

We use $\psi$ and $\{\psi_j\}_{j=1}^m$ to denote the test question of interest and the $m$ similar ones, respectively. Students' performance on test questions is characterized by a *performance function* $F$: a student with profile $z \in \mathcal{Z}$ answers a question $\psi$ correctly with probability $F(z, \psi) \in [0, 1]$. The average student performance on the test questions $\psi$ and $\{\psi_j\}_{j=1}^m$ are then $\mu = \mathbb{E}_{z \sim \mathcal{P}} F(z, \psi)$ and $\mu_j = \mathbb{E}_{z \sim \mathcal{P}} F(z, \psi_j)$, respectively. In addition, the LLM generates synthetic student performance from a *synthetic performance function* $F^{\mathsf{syn}}$: when prompted with a synthetic profile $z^{\mathsf{syn}} \in \mathcal{Z}$, the LLM answers a question $\psi$ correctly with probability $F^{\mathsf{syn}}(z^{\mathsf{syn}}, \psi) \in [0, 1]$.

The collection of the real dataset $\mathcal{D}_j = \{y_{j,i}\}_{i=1}^{n_j}$ can be thought of as drawing $n_j$ i.i.d. student profiles $\{z_{j,i}\}_{i=1}^{n_j} \sim \mathcal{P}$ and then sampling $y_{j,i} \sim \text{Bernoulli}(F(z_{j,i}, \psi_j))$ for each $i \in [n_j]$. Similarly, the generation of the synthetic dataset $\mathcal{D}_j^{\mathsf{syn}} = \{y_{j,i}^{\mathsf{syn}}\}_{i=1}^K$ can be thought of as drawing i.i.d. synthetic profiles $\{z_{j,i}^{\mathsf{syn}}\}_{i=1}^K \sim \mathcal{P}^{\mathsf{syn}}$ and then sampling $y_{j,i}^{\mathsf{syn}} \sim \text{Bernoulli}(F^{\mathsf{syn}}(z_{j,i}^{\mathsf{syn}}, \psi_j))$ for each $i \in [K]$. For $\mathcal{D}^{\mathsf{syn}} = \{y_i^{\mathsf{syn}}\}_{i=1}^K$, we adopt a similar notation $\{z_i^{\mathsf{syn}}\}_{i=1}^K$ for the synthetic profiles. We note that when collecting real or synthetic samples, the performance functions never appear explicitly. They are introduced only to facilitate the problem formulation.

Finally, we assume that the test questions are drawn randomly from a question bank, and that the datasets are independent.

**Assumption 2.2** (Randomly sampled questions)**.** The questions $\psi, \psi_1, ..., \psi_m$ are independently sampled from a distribution over a space $\Psi$.

**Assumption 2.3** (Independent data)**.** For each $j \in [m]$, conditioned on $\psi_j$, the datasets $\mathcal{D}_j$ and $\mathcal{D}_j^{\mathsf{syn}}$ are independent. Conditioned on $\psi_1, ..., \psi_m$, the dataset tuples $(\mathcal{D}_1, \mathcal{D}_1^{\mathsf{syn}}), ..., (\mathcal{D}_m, \mathcal{D}_m^{\mathsf{syn}})$ are independent. Finally, $(\psi, \mathcal{D}^{\mathsf{syn}})$ is independent of $\{(\psi_j, \mathcal{D}_j, \mathcal{D}_j^{\mathsf{syn}})\}_{j=1}^m$.

We are now ready to state the theoretical guarantee of our approach. Its proof is deferred to Appendix B.2. We note that the assumption $\mathbb{P}(\bar{y}_j \leq \mu_j \mid \psi_j) = 1/2$ is a CLT approximation and is for mathematical convenience only.

**Theorem 2.4** (Coverage guarantee)**.** *Let Assumptions 2.2*

and 2.3 hold. Assume that $\mathbb{P}(\bar{y}_j \leq \mu_j \mid \psi_j) = 1/2$ for each $j \in [m]$. Fix $\alpha \in (0,1)$. Then the simulation sample size $\widehat{k}$ defined by (7) satisfies

$$\mathbb{P}\Big(\mu \in \mathcal{I}^{\mathsf{syn}}(\widehat{k})\Big) \geq 1 - \alpha - \sqrt{\frac{2}{m}}.$$

The probability is taken with respect to the randomness of $\big\{(\psi_j, \mathcal{D}_j, \mathcal{D}_j^{\mathsf{syn}})\big\}_{j=1}^m$, $\psi$ and $\mathcal{D}^{\mathsf{syn}}$.

On average, the chosen simulation sample size $\widehat{k}$ leads to a confidence interval $\mathcal{I}^{\mathsf{syn}}(\widehat{k})$ that covers the true mean $\mu$ with probability at least $1 - \alpha - O(\sqrt{1/m})$. As $m \to \infty$, the aforementioned lower bound converges to $1 - \alpha$.

### 2.4. Sharpness of Sample Size Selection

We have seen that the chosen interval $\mathcal{I}^{\mathsf{syn}}(\widehat{k})$ has good coverage properties. In this section, we complement this result by showing that the interval is not overly conservative. To simplify computation, we slightly modify the setting.

**Example 2.5** (Gaussian performance score). Consider the setting in Section 2.3 with the following modifications. On a test question $\psi \in \Psi$, the performance (e.g., score) of a real student follows a Gaussian distribution with mean $\mathbb{E}_{z \sim \mathcal{P}} F(z, \psi)$ and variance 1, instead of a Bernoulli distribution with mean $\mathbb{E}_{z \sim \mathcal{P}} F(z, \psi)$. Similarly, the performance of the LLM follows a Gaussian distribution with mean $\mathbb{E}_{z^{\mathsf{syn}} \sim \mathcal{P}^{\mathsf{syn}}} F^{\mathsf{syn}}(z^{\mathsf{syn}}, \psi)$ and variance 1. Moreover, the confidence intervals $\mathcal{I}^{\mathsf{syn}}(k)$ defined in (3) and $\mathcal{I}_j^{\mathsf{syn}}(k)$ defined in (4) are changed to

$$\mathcal{I}^{\mathsf{syn}}(k) = \left[ \bar{y}_k^{\mathsf{syn}} - \frac{C}{\sqrt{k}}, \ \bar{y}_k^{\mathsf{syn}} + \frac{C}{\sqrt{k}} \right],$$

$$\mathcal{I}_j^{\mathsf{syn}}(k) = \left[ \bar{y}_{j,k}^{\mathsf{syn}} - \frac{C}{\sqrt{k}}, \ \bar{y}_{j,k}^{\mathsf{syn}} + \frac{C}{\sqrt{k}} \right],$$

respectively, where $C = 2\Phi^{-1}(1 - \alpha/4)$. For simplicity, we suppose that the real datasets have the same size: $n_j = n$ for all $j \in [m]$. Finally, we define

$$\Delta = \sup_{\psi \in \Psi} \big| \mathbb{E}_{z \sim \mathcal{P}} F(z, \psi) - \mathbb{E}_{z^{\mathsf{syn}} \sim \mathcal{P}^{\mathsf{syn}}} F^{\mathsf{syn}}(z^{\mathsf{syn}}, \psi) \big|.$$

In Example 2.5, the quantity $\Delta$ measures the discrepancy between the distributions of the real students' performance and of the simulated students' performance. The following theorem presents a lower bound on the chosen simulation sample size $\widehat{k}$. Its proof can be found in Appendix B.3.

**Theorem 2.6** (Sharpness of chosen sample size). *Consider the setting of Example 2.5. Let $\widehat{k}$ be chosen by the procedure* (7). *Choose $\delta \in (0,1)$. There exists a constant $C' > 0$ determined by $\alpha$ such that when $m > C' \log(n/\delta)$, the*

following holds with probability at least $1 - \delta$:

$$\widehat{k} \geq \min\left\{ K, \ n, \ \left(\frac{C}{5\Delta}\right)^2 \right\}.$$

When this happens, the selected confidence interval $\mathcal{I}^{\mathsf{syn}}(\widehat{k})$ has width $O\left(\max\{K^{-1/2}, n^{-1/2}, \Delta\}\right)$.

Theorem 2.6 implies that the interval $\mathcal{I}^{\mathsf{syn}}(\widehat{k})$ is the shortest possible. To see this, suppose that the simulation budget $K$ is large, then with high probability, $\mathcal{I}^{\mathsf{syn}}(\widehat{k})$ has width $O(\max\{\Delta, n^{-1/2}\})$. This is the optimal width because of the following reasons. First, in the worst case, any $\mathcal{I}^{\mathsf{syn}}(k)$ that covers the true mean with high probability must have width $\Omega(\Delta)$ in order to address the distribution shift between the real and simulated responses. Second, as $n$ real human responses can identify the true mean up to an error of $O(n^{-1/2})$, then any valid $\mathcal{I}^{\mathsf{syn}}(k)$ must also have width $\Omega(n^{-1/2})$. This shows the sharpness of the chosen sample size $\widehat{k}$ and the confidence interval $\mathcal{I}^{\mathsf{syn}}(\widehat{k})$.

## 3. General Setup and Methodology

In this section, we study the more general setting where survey responses and confidence sets can be multi-dimensional.

### 3.1. Problem Formulation

Let $\mathcal{Z}$ be a profile space, $\mathcal{P}$ a probability distribution over $\mathcal{Z}$ which represents the true population, and $\mathcal{P}^{\mathsf{syn}}$ a synthetic distribution over $\mathcal{Z}$ used to generate synthetic profiles.

Let $\Psi$ be a collection of survey questions, and $\mathcal{Y}$ be the space of possible responses to the survey questions. When a person with profile $z \in \mathcal{Z}$ is asked a survey question $\psi \in \Psi$, the person gives a response $y$ following a distribution $\mathcal{Q}(\cdot \mid z, \psi)$ over $\mathcal{Y}$. We are interested in the distribution of the population's response to the survey question $\psi$, which is given by $\mathcal{R}(\cdot \mid \psi) = \int_{\mathcal{Z}} \mathcal{Q}(\cdot \mid z, \psi) \mathcal{P}(dz)$. In particular, we seek to construct a confidence set for some statistic $\theta(\psi)$ of $\mathcal{R}(\cdot \mid \psi)$, which can be multi-dimensional, say in $\mathbb{R}^d$. Below we revisit the educational test example in Section 2 in this framework. More examples are provided in Appendix A.

**Example 3.1** (Educational test evaluation). In the educational test evaluation example in Section 2, each $z \in \mathcal{Z}$ is a student profile, each $\psi \in \Psi$ is a test question, the response space is $\mathcal{Y} = \{0, 1\}$, and $\mathcal{Q}(\cdot \mid z, \psi) = \mathrm{Bernoulli}(F(z, \psi))$. The statistic $\theta(\psi)$ is the probability of a student answering the question correctly: $\mathbb{E}_{y \sim \mathcal{R}(\cdot \mid \psi)}[y] = \mathbb{E}_{z \sim \mathcal{P}}[F(z, \psi)]$.

We consider constructing the confidence set by using simulated responses from an LLM. Given a profile $z$, a survey question $\psi$ and a prompt $p$, the LLM simulates a response

$y^{\mathsf{syn}}$ from a distribution $\mathcal{Q}^{\mathsf{syn}}(\,\cdot\mid z,\psi,p)$ which aims to mimic $\mathcal{Q}(\,\cdot\mid z,\psi)$. We can generate i.i.d. synthetic profiles $\{z_i^{\mathsf{syn}}\}_{i=1}^{K}\sim\mathcal{P}^{\mathsf{syn}}$, then feed them into the LLM along with $\psi$ and $p$. The LLM then generates synthetic responses $\{y_i^{\mathsf{syn}}\}_{i=1}^{K}$, where $y_i^{\mathsf{syn}}\sim\mathcal{Q}^{\mathsf{syn}}(\,\cdot\mid z_i^{\mathsf{syn}},\psi,p)$. Here $K\in\mathbb{Z}_+$ is the simulation budget.

Using the simulated samples $\mathcal{D}^{\mathsf{syn}}=\{y_i^{\mathsf{syn}}\}_{i=1}^{K}$, we can construct a family of candidate confidence sets such as the one-dimensional CLT-based confidence interval (3). More generally, the statistics literature has developed a variety of approaches such as inverting hypothesis tests (Casella & Berger, 2002), the bootstrap (Efron, 1979), and the empirical likelihood ratio function (Owen, 1990). We will assume access to a black-box procedure $\mathcal{C}$ that takes as input a dataset $\mathcal{D}$ and outputs a confidence set $\mathcal{C}(\mathcal{D})\subseteq\mathbb{R}^d$. Then, we can construct a family of confidence sets $\{\mathcal{S}^{\mathsf{syn}}(k)\}_{k=1}^{K}$ by

$$\mathcal{S}^{\mathsf{syn}}(k)=\mathcal{C}\left(\{y_i^{\mathsf{syn}}\}_{i=1}^{k}\right). \tag{9}$$

We also set $\mathcal{S}^{\mathsf{syn}}(0)=\mathbb{R}^d$, so $\theta(\psi)\in\mathcal{S}^{\mathsf{syn}}(0)$ always. We will not impose any assumptions on the quality of the confidence sets produced by $\mathcal{C}$.

As the LLM may not be a faithful reflection of the true human population, we will make use of real data to choose a good confidence set from $\{\mathcal{S}^{\mathsf{syn}}(k)\}_{k=1}^{K}$. We assume that we have collected real human responses from $m$ surveys $\psi_1,...,\psi_m\in\Psi$. For each $j\in[m]$, we have responses $\mathcal{D}_j=\{y_{j,i}\}_{i=1}^{n_j}$ from $n_j$ i.i.d. surveyees $\{z_{j,i}\}_{i=1}^{n_j}\sim\mathcal{P}$, with $y_{j,i}\sim\mathcal{Q}(\,\cdot\mid z_{j,i},\psi_j)$.

We also simulate LLM responses to these $m$ survey questions. For each $j\in[m]$, we feed i.i.d. synthetic profiles $\{z_{j,i}^{\mathsf{syn}}\}_{i=1}^{K}\sim\mathcal{P}^{\mathsf{syn}}$, the question $\psi_j$ and the prompt $p$ into the LLM, which then simulates responses $\mathcal{D}_j^{\mathsf{syn}}=\{y_{j,i}^{\mathsf{syn}}\}_{i=1}^{K}$ with $y_{j,i}^{\mathsf{syn}}\sim\mathcal{Q}^{\mathsf{syn}}(\,\cdot\mid z_{j,i}^{\mathsf{syn}},\psi_j,p)$. The datasets $\{\mathcal{D}_j\}_{j=1}^{m}$ and $\{\mathcal{D}_j^{\mathsf{syn}}\}_{j=1}^{m}$ will be used to select a confidence set from $\{\mathcal{S}^{\mathsf{syn}}(k)\}_{k=1}^{K}$. We make the same Assumptions 2.2 and 2.3 as in Section 2.

We are now ready to formally state our problem.

**Problem 3.2** (Uncertainty quantification). Given $\alpha\in(0,1)$, how to use $\{\mathcal{D}_j\}_{j=1}^{m}$ and $\{\mathcal{D}_j^{\mathsf{syn}}\}_{j=1}^{m}$ to choose $\widehat{k}\in[K]$ such that

$$\mathbb{P}\left(\theta(\psi)\in\mathcal{S}^{\mathsf{syn}}(\widehat{k})\right)\approx 1-\alpha?$$

### 3.2. General Methodology for Sample Size Selection

We now present our general methodology for Problem 3.2. For each $j\in[m]$, we form confidence sets similar to (9) using the synthetic data $\mathcal{D}_j^{\mathsf{syn}}$:

$$\mathcal{S}_j^{\mathsf{syn}}(k)=\mathcal{C}\left(\{y_{j,i}^{\mathsf{syn}}\}_{i=1}^{k}\right),\quad\forall k\in[K]. \tag{10}$$

We also set $\mathcal{S}_j^{\mathsf{syn}}(0)=\mathbb{R}^d$. We will pick $\widehat{k}\in\{0,1,...,K\}$ such that $\mathcal{S}_j^{\mathsf{syn}}(\widehat{k})$ is a good confidence interval for $\theta(\psi_j)$ for

each $j\in[m]$. This choice of $\widehat{k}$ will also be good for $\mathcal{S}^{\mathsf{syn}}(k)$, thanks to the i.i.d. assumption on the survey questions.

Ideally, we would like to pick $k$ such that $(1-\alpha)$ coverage is achieved empirically over the $m$ survey questions:

$$\frac{1}{m}\sum_{j=1}^{m}\mathbf{1}\{\theta(\psi_j)\notin\mathcal{S}_j^{\mathsf{syn}}(k)\}\leq\alpha. \tag{11}$$

However, the population-level quantities $\{\theta(\psi_j)\}_{j=1}^{m}$ are not available, so we must approximate them by the real data $\{\mathcal{D}_j\}_{j=1}^{m}$. In Section 2, we have taken the approach of constructing unbiased point estimates, but it does not directly extend to the more general case.

Instead, we will use the real data $\{\mathcal{D}_j\}_{j=1}^{m}$ to construct confidence sets for $\{\theta(\psi_j)\}_{j=1}^{m}$. Choose a confidence level $\gamma\in(0,1)$. For each $j\in[m]$, we use $\mathcal{D}_j$ to construct a confidence set $\mathcal{S}_j$ that satisfies

$$\mathbb{P}\left(\theta(\psi_j)\in\mathcal{S}_j\,\Big|\,\psi_j\right)\geq\gamma. \tag{12}$$

These confidence sets are easy to construct as the samples in $\mathcal{D}_j$ follow the true response distribution. When $\theta(\psi_j)\in\mathcal{S}_j$, the condition $\mathcal{S}_j\subseteq\mathcal{S}_j^{\mathsf{syn}}(k)$ is sufficient for $\theta(\psi_j)\in\mathcal{S}_j^{\mathsf{syn}}(k)$. Equivalently, when $\theta(\psi_j)\in\mathcal{S}_j$, the condition $\theta(\psi_j)\notin\mathcal{S}_j^{\mathsf{syn}}(k)$ must imply $\mathcal{S}_j\nsubseteq\mathcal{S}_j^{\mathsf{syn}}(k)$. Thus, we take

$$L(k)=\frac{1}{m}\sum_{j=1}^{m}\mathbf{1}\{\mathcal{S}_j\nsubseteq\mathcal{S}_j^{\mathsf{syn}}(k)\} \tag{13}$$

as a proxy for the empirical miscoverage. Since $\theta(\psi_j)\in\mathcal{S}_j$ with probability $\gamma$, then the frequency of having $\mathcal{S}_j\nsubseteq\mathcal{S}_j^{\mathsf{syn}}(k)$ is at least $\gamma$ times the frequency of $\theta(\psi_j)\notin\mathcal{S}_j^{\mathsf{syn}}(k)$. Roughly speaking,

$$L(k)\geq\gamma\cdot\left(\frac{1}{m}\sum_{j=1}^{m}\mathbf{1}\{\theta(\psi_j)\notin\mathcal{S}_j^{\mathsf{syn}}(k)\}\right). \tag{14}$$

Combining (11) and (14) leads to the following criterion for selecting $k$:

$$\widehat{k}=\max\left\{0\leq k\leq K:\,L(i)\leq\gamma\alpha,\,\forall i\leq k\right\}. \tag{15}$$

Note that $\widehat{k}$ is well-defined because $L(0)=0$. The full procedure for sample size selection is summarized in Algorithm 1.

We now present the coverage guarantee for our method, which shows that the chosen confidence set $\mathcal{S}^{\mathsf{syn}}(\widehat{k})$ has coverage probability at least $1-\alpha-O(1/\sqrt{m})$. Its proof is deferred to Appendix B.4.

**Theorem 3.3.** *Let Assumptions 2.2 and 2.3 hold. Fix $\alpha\in(0,1)$. The output $\widehat{k}$ of Algorithm 1 satisfies*

$$\mathbb{P}\left(\theta(\psi)\in\mathcal{S}^{\mathsf{syn}}(\widehat{k})\right)\geq 1-\alpha-\gamma^{-1}\sqrt{\frac{1}{2m}}.$$

**Algorithm 1** Simulation Sample Size Selection

**Input:** Survey questions with real and simulated responses $\{(\psi_j, \mathcal{D}_j, \mathcal{D}_j^{\mathsf{syn}})\}_{j=1}^{m}$, prescribed miscoverage probability $\alpha$, confidence set construction procedure $\mathcal{C}$, confidence level $\gamma$, simulation budget $K$.

**for** $j = 1, ..., m$ **do**
    Use $\mathcal{C}$ and $\mathcal{D}_j^{\mathsf{syn}}$ to construct synthetic confidence sets $\{\mathcal{S}_j^{\mathsf{syn}}(k)\}_{k=0}^{K}$ by (10).
    Use $\mathcal{D}_j$ to construct a confidence set $\mathcal{S}_j$ satisfying (12).
**end for**
Define

$$L(k) = \frac{1}{m} \sum_{j=1}^{m} \mathbf{1}\{\mathcal{S}_j \not\subseteq \mathcal{S}_j^{\mathsf{syn}}(k)\}.$$

Set $\widehat{k} = \max\{0 \leq k \leq K : L(i) \leq \gamma\alpha, \ \forall i \leq k\}$.

**Output:** Sample size $\widehat{k}$.

It is worth noting that our method achieves this coverage without any assumptions on the qualities of the LLM and the procedure $\mathcal{C}$ for confidence set construction. Nevertheless, the size of the chosen confidence set $\mathcal{S}^{\mathsf{syn}}(\widehat{k})$ depends on these factors. If there is a large alignment gap between the LLM and the human population, then $\mathcal{S}^{\mathsf{syn}}(\widehat{k})$ will inevitably be large.

## 4. Numerical Experiments

In this section, we apply our general method in Section 3 to LLMs over real datasets. The code and data are available at https://github.com/yw3453/uq-llm-survey-simulation.

### 4.1. Experiment Setup

**LLMs.** We consider 8 LLMs: GPT-3.5-Turbo (gpt-3.5-turbo), GPT-4o (gpt-4o), and GPT-4o-mini (gpt-4o-mini) (OpenAI, 2022; 2024a;b); Claude 3.5 Haiku (claude-3-5-haiku-20241022) (Anthropic, 2024); Llama 3.1 8B (Llama-3-8B-Instruct-Turbo) and Llama 3.3 70B (Llama-3.3-70B-Instruct-Turbo) (Dubey et al., 2024); Mistral 7B (Mistral-7B-Instruct-v0.3) (Jiang et al., 2023); DeepSeek-V3 (DeepSeek-V3) (Liu et al., 2024).

**Datasets.** We use two datasets for survey questions, each corresponding to one uncertainty quantification task. The first dataset is the OpinionQA dataset (Santurkar et al., 2023). It was built from Pew Research's American Trends

Panel[1], and contains the general US population's responses to survey questions spanning topics such as science, politics, and health. After pre-processing we have 385 unique questions and 1,476,868 responses to these questions from at least 32,864 people. These questions have 5 choices corresponding to ordered sentiments which we map to sentiment scores $-1, -\frac{1}{3}, 0, \frac{1}{3}, 1$. Each question has at least 400 responses. For each response, we have information on the surveyee's political profile, religious affiliation, educational background, socio-economic status, etc., which we use to generate synthetic profiles. See Appendix C.1 for more details. We consider the task of constructing a confidence interval for the US population's average sentiment score for a survey question. This is the setup in Example A.3.

The second dataset is the EEDI dataset created by (He-Yueya et al., 2024), which was built upon the NeurIPS 2020 Education Challenge dataset (Wang et al., 2021). It consists of students' responses to mathematics multiple-choice questions on the Eedi online educational platform[2]. The dataset contains 573 unique questions and 443,433 responses to these questions from 2,287 students. All questions have four choices (A, B, C, D). Out of these questions, we use questions that have at least 100 student responses. Excluding questions with graphs or diagrams, we are left with a total of 412 questions. For each student, we have information on their gender, age, and socioeconomic status, which we use to generate synthetic profiles. See Appendix C.2 for more details. We consider the task of constructing a confidence interval for the probability of a student answering a question correctly. This is similar to the setup in Example 3.1.

**Confidence set construction.** Our method can be built upon any arbitrary confidence set construction procedure $\mathcal{C}$. We use a construction procedure based on Hoeffding's concentration inequality (e.g., Theorem 2.8 in (Boucheron et al., 2013)), as it has valid coverage guarantee for any finite sample size. Given $\alpha \in (0, 1)$ and responses $\{y_i^{\mathsf{syn}}\}_{i=1}^{k}$, we construct the confidence interval

$$\mathcal{C}(\{y_i^{\mathsf{syn}}\}_{i=1}^{k}) = \left[ \bar{y}_k^{\mathsf{syn}} - cM\sqrt{\frac{\log(2/\alpha)}{2k}}, \right.$$
$$\left. \bar{y}_k^{\mathsf{syn}} + cM\sqrt{\frac{\log(2/\alpha)}{2k}} \right], \quad (16)$$

where $\bar{y}_k^{\mathsf{syn}} = \frac{1}{k}\sum_{i=1}^{k} y_i^{\mathsf{syn}}$ is the sample mean, $M > 0$ is an upper bound on the range of the responses, and $c > 1$ is a scaling constant.

**Hyperparameters.** We consider $\alpha \in \{0.05 \cdot \ell : \ell \in [10]\}$, $c = \sqrt{2}$ and $\gamma = 0.5$. For the EEDI dataset, we set the

---

[1] https://www.pewresearch.org/the-american-trends-panel/
[2] https://eedi.com/

simulation budget $K = 50$ and take $M = 1$ since the responses are binary. For the OpinionQA dataset, we set the simulation budget $K = 100$ and take $M = 2$ since the responses range within $[-1, 1]$.

## 4.2. Experiment Procedure

We now describe our experiment procedure for applying the method in Section 3 to each dataset. Denote the dataset by $\{(\psi_j, \mathcal{D}_j)\}_{j=1}^J$, where $\psi_j$ is a survey question and $\mathcal{D}_j = \{y_{j,i}\}_{i=1}^{n_j}$ is a collection of human responses. For each $j \in [J]$, we simulate $K$ responses $\mathcal{D}_j^{\mathsf{syn}}$ from an LLM. We then randomly split $\mathscr{D} = \{(\mathcal{D}_j, \mathcal{D}_j^{\mathsf{syn}})\}_{j=1}^J$ into a training set $\mathscr{D}^{\mathrm{tr}} = \{(\mathcal{D}_j, \mathcal{D}_j^{\mathsf{syn}})\}_{j \in \mathcal{J}_{\mathrm{tr}}}$ and a testing set $\mathscr{D}^{\mathrm{te}} = \{(\mathcal{D}_j, \mathcal{D}_j^{\mathsf{syn}})\}_{j \in \mathcal{J}_{\mathrm{te}}}$, with $|\mathscr{D}^{\mathrm{tr}}| : |\mathscr{D}^{\mathrm{te}}| = 3 : 2$.

**Selection of simulation sample size.** We apply the approach (15) with the training set $\mathscr{D}^{\mathrm{tr}}$ to select a simulation sample size $\widehat{k}$. For the confidence set $\mathcal{S}_j$ in (12) constructed from the real data $\mathcal{D}_j$, we use the standard CLT-based confidence interval:

$$\mathcal{S}_j = \left[ \bar{y}_j - \frac{s_j}{\sqrt{n_j}} \Phi^{-1}\left(\frac{1+\gamma}{2}\right), \right.$$
$$\left. \bar{y}_j + \frac{s_j}{\sqrt{n_j}} \Phi^{-1}\left(\frac{1+\gamma}{2}\right) \right], \quad (17)$$

where $\bar{y}_j = \frac{1}{n_j} \sum_{i=1}^{n_j} y_{j,i}$ and $s_j = \sqrt{\bar{y}_j(1 - \bar{y}_j)}$. Since $n_j$ is at least 100, $\mathcal{S}_j$ has approximately $\gamma$ coverage probability. In Figure 5 of Appendix C.3, we provide a visualization for the selection of $\widehat{k}$.

**Evaluation of selected sample size.** We use $\mathscr{D}^{\mathrm{te}}$ to evaluate the quality of the chosen simulation sample size $\widehat{k}$. As the true population parameter $\theta(\psi)$ is unavailable, the true coverage probability $\mathbb{P}(\theta(\psi) \in \mathcal{S}^{\mathsf{syn}}(\widehat{k}))$ cannot be computed. However, we can apply the same idea as (13) in Section 3 to compute a proxy for the miscoverage level. For each survey question $j \in \mathcal{J}_{\mathrm{te}}$, the selected sample size $\widehat{k}$ leads to the synthetic confidence set $\mathcal{S}_j^{\mathsf{syn}}(\widehat{k}) = \mathcal{C}(\{y_{j,i}^{\mathsf{syn}}\}_{i=1}^{\widehat{k}})$. We form the confidence set $\mathcal{S}_j$ from real data $\mathcal{D}_j$ as in (17) and define

$$\widetilde{L}(k) = \frac{1}{\gamma} \cdot \frac{1}{|\mathcal{J}_{\mathrm{te}}|} \sum_{j \in \mathcal{J}_{\mathrm{te}}} \mathbf{1}\left\{ \mathcal{S}_j \not\subseteq \mathcal{S}_j^{\mathsf{syn}}(k) \right\}. \quad (18)$$

The proof of Theorem 3.3 shows that, for every $k \in [K]$ and survey question $j$,

$$\gamma \cdot \mathbb{P}(\theta(\psi) \notin \mathcal{S}^{\mathsf{syn}}(k)) \leq \mathbb{P}(\mathcal{S}_j \not\subseteq \mathcal{S}_j^{\mathsf{syn}}(k)) = \gamma \cdot \mathbb{E}[\widetilde{L}(k)].$$

Thus, if $\mathbb{E}[\widetilde{L}(\widehat{k})] \leq \alpha$, then $\mathbb{P}(\theta(\psi) \notin \mathcal{S}^{\mathsf{syn}}(\widehat{k})) \leq \alpha$ must hold. To this end, we will test a hypothesis $H_0 : \mathbb{E}[\widetilde{L}(\widehat{k})] \leq \alpha$ against its alternative $H_1 : \mathbb{E}[\widetilde{L}(\widehat{k})] > \alpha$.

## 4.3. Experiment Results

For both datasets, we evaluate three metrics as $\alpha$ varies: the miscoverage probability proxy (18), the selected simulation sample size $\widehat{k}$, and the half-width of the synthetic confidence interval $\mathcal{S}^{\mathsf{syn}}(\widehat{k})$. We consider 100 random train-test splits of the questions. For compactness, we present results on the miscoverage probability proxy and $\widehat{k}$, and defer the results on the half-width of the synthetic confidence interval as well as more experiment details to Appendix D. We omit Llama 3.1 8B for the EEDI dataset experiment because it frequently failed to answer EEDI questions in required formats. As a baseline, we also include a naïve response generator (random) that chooses an available answer uniformly at random.

**Coverage validity.** In Figure 2, we present histograms of $p$-values for the hypothesis test $\mathbb{E}[\widetilde{L}(\widehat{k})] \leq \alpha$ against $\mathbb{E}[\widetilde{L}(\widehat{k})] > \alpha$ across various LLMs and $\alpha$'s over the OpinionQA and EEDI datasets. The $p$-values are computed using a one-sided $z$-test over the 100 random splits. As can be seen from the histograms, all $p$-values are reasonably large, indicating that the hypothesis $\mathbb{E}[\widetilde{L}(\widehat{k})] \leq \alpha$ cannot be rejected (e.g., at the 0.05 significance level) for any LLM and $\alpha$ across both datasets. These experiment results verify the theoretical guarantees in Section 3, showing that the miscoverage rate is effectively controlled by our method.

**Selected simulation sample size.** In Figure 3, we plot the average $\widehat{k}$ over the 100 random splits for various LLMs on the OpinionQA and EEDI datasets, respectively. The error bars represent 95% confidence intervals.

In general, a larger $\widehat{k}$ means that the LLM has a stronger simulation power. On the OpinionQA dataset, GPT-4o has the best performance. On the EEDI dataset, DeepSeek-V3 has the best performance, followed by GPT-4o and Claude 3.5 Haiku. Interestingly, on the OpinionQA dataset all LLMs clearly outperform the random benchmark, while on the EEDI dataset only DeepSeek-V3 and GPT-4o seem to outperform the random benchmark. Moreover, LLMs exhibit uniformly higher $\widehat{k}$ on the OpinionQA dataset than on the EEDI dataset, suggesting higher fidelity in simulating subjective opinions to social problems than in simulating student answers to mathematics questions.

The experiment results demonstrate the importance of a disciplined approach to using synthetic samples. The ease of LLM-based simulation makes it tempting to generate a large number of responses per question. However, our results show that there is great heterogeneity in the simulation powers of different LLMs over different datasets: the largest $\widehat{k}$ is below 100, while the smallest $\widehat{k}$ could be in the single digits. This means that there is real peril in using too many

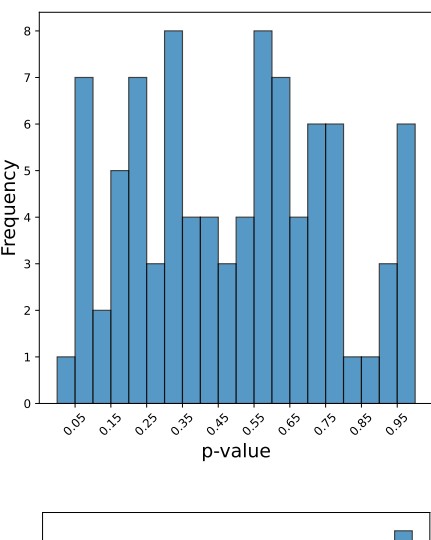

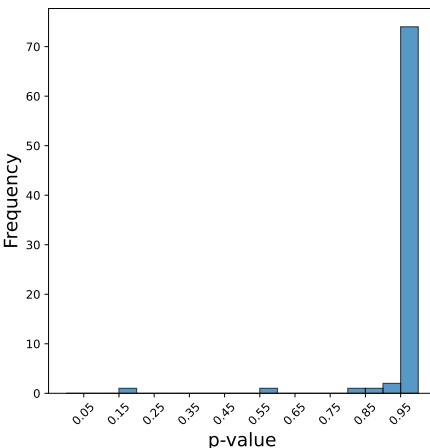

Figure 2. Histograms of $p$-values for the hypothesis test $\mathbb{E}\big[\widetilde{L}(\widehat{k})\big] \leq \alpha$ against $\mathbb{E}\big[\widetilde{L}(\widehat{k})\big] > \alpha$ across various LLMs and $\alpha$'s over the OpinionQA dataset (top) and the EEDI dataset (bottom).

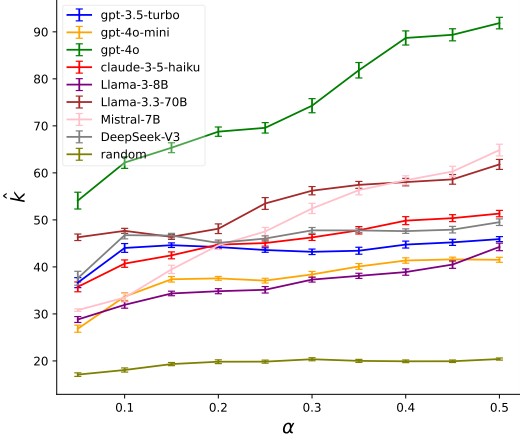

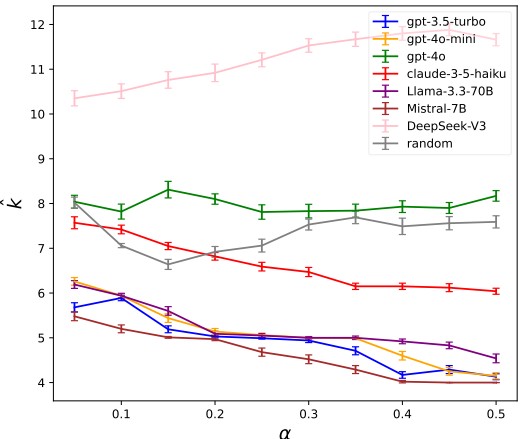

Figure 3. Average $\widehat{k}$ for various LLMs and $\alpha$ over the OpinionQA dataset (top) and the EEDI dataset (bottom).

synthetic samples and being overly confident in the results.

## 5. Discussions

We developed a general approach for converting imperfect LLM-based survey simulations into statistically valid confidence sets for population parameters of human responses. It identifies a simulation sample size which is useful for future simulation tasks and which quantifies the fidelity of LLM simulations. Numerical experiments on real datasets verified the coverage guarantees of our approach, and revealed that existing LLMs exhibited higher fidelity in simulating opinions to social problems than in simulating student answers to mathematics questions.

Several future directions are worth exploring. First, our approach does not explicitly minimize the size of the prediction set. A natural question is whether we can incorporate a size minimization procedure to produce smaller confidence

sets with good coverage. Second, it would be interesting to see if our approach can be combined with debiasing methods to give more informative confidence sets. Finally, as prompt engineering is known to have crucial effects on the quality of LLM generations, it is worth investigating the impacts of prompts on the selected simulation sample size $\widehat{k}$, and how prompt engineering can be leveraged to improve the fidelity of LLM simulations.

## Acknowledgements

Chengpiao Huang and Kaizheng Wang's research is supported by an NSF grant DMS-2210907 and a Data Science Institute seed grant SF-181 at Columbia University.

## Impact Statement

This work develops methodologies for uncertainty quantification for LLM-simulated survey responses. We believe that it can facilitate more reliable LLM-simulated experi-

ments in areas such as social science, economics, marketing, and education. On the other hand, we point out that the theoretical guarantees for our method hold only on average over the randomness of the survey questions and the real and simulation responses, and do not necessarily hold for every random instance. Therefore, we recommend caution when applying the confidence intervals to decision-making.

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

# A. Illustrations and Examples

We first present a visualization of the relation between the simulation sample size and the LLM's simulation power in Figure 4.

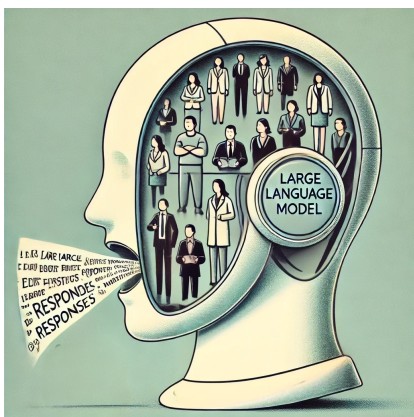

*Figure 4.* An interpretation of the simulation sample size $\widehat{k}$ as the size of human population that an LLM can represent. Generating outputs from the LLM can be thought of as "resampling" from $\widehat{k}$ human samples that make up the LLM. The figure is generated by DALL-E (Ramesh et al., 2021), and borrows ideas from the *Mechanical Turk*, a chess-playing machine from the 18th century with a human player hidden inside.

Below we provide more examples for the general problem framework in Section 3. The survey responses can be real-valued or multi-dimensional.

**Example A.1** (Market research). Suppose a company is interested in learning its customers' willingness-to-pay (WTP) for a new product, which is the highest price a customer is willing to pay for the product. Then, each $z \in \mathcal{Z}$ can represent a customer profile (e.g., age, gender, occupation), each survey question $\psi$ is about a certain product, and a customer's response $y$ is a noisy observation of the customer's WTP. Then $\mathcal{R}(\,\cdot\mid\psi)$ is the distribution of the customer population's WTP. We may take $\theta(\psi)$ as the $\tau$-quantile of the WTP distribution $\mathcal{R}(\,\cdot\mid\psi)$, for some $\tau \in (0, 1)$:

$$\theta(\psi) = \inf\left\{q \in [0, \infty) : \mathbb{P}_{y\sim\mathcal{R}(\cdot\mid\psi)}(y \leq q) \geq \tau\right\}.$$

An LLM can be used to simulate customers' WTP for the product.

**Example A.2** (Public survey, multi-dimensional). Suppose an organization is interested in performing a public survey in a city. Each survey question $\psi$ is a multiple-choice question with 5 options. An example is "How often do you talk to your neighbors?", with 5 choices "Basically every day", "A few times a week", "A few times a month", "Once a month", and "Less than once a month". Every $z \in \mathcal{Z}$ is a person's profile (e.g., age, gender, occupation), the response space $\mathcal{Y}$ is the standard orthonormal basis $\{e_i\}_{i=1}^5$ in $\mathbb{R}^5$, where $y = e_i$ indicates that a person chooses the $i$-th option. We can take $\theta(\psi) = \mathbb{E}_{y\sim\mathcal{Q}(\cdot\mid\psi)}[y] \in \mathbb{R}^5$, which summarizes the proportion of people that choose each option. An LLM can be used to simulate people's answers to the survey question.

**Example A.3** (Public survey, one-dimensional). Consider the setup in Example A.2. When the 5 choices in a survey question correspond to ordered sentiments, we can map them to numeric scores, say, $v = (-1, -\frac{1}{3}, 0, \frac{1}{3}, 1)^\top$. Then the statistic $\widetilde{\theta}(\psi) = \langle v, \theta(\psi)\rangle$ reflects the population's average sentiment for the survey question $\psi$.

# B. Proofs

## B.1. Failure of Exact CLT-Based Intervals under Distribution Shift

Consider a true distribution $N(\mu, 1)$ and a synthetic distribution $N(\mu^{\mathsf{syn}}, 1)$. Suppose we draw $k$ i.i.d. synthetic samples $\{x_i\}_{i=1}^k \sim N(\mu^{\mathsf{syn}}, 1)$ and construct the standard $(1 - \alpha)$ confidence interval for $\mu$:

$$\mathcal{I}^{\mathsf{syn}}(k) = \left[\bar{x}_k - \frac{\Phi^{-1}(1 - \alpha/2)}{\sqrt{k}}, \ \bar{x}_k + \frac{\Phi^{-1}(1 - \alpha/2)}{\sqrt{k}}\right],$$

where $\bar{x}_k = \frac{1}{k}\sum_{i=1}^{k} x_i$. Let $\Delta = \mu - \mu^{\text{syn}}$, which represents the discrepancy between the true distribution and the synthetic distribution. Then

$$\mathbb{P}\Big(\mu \in \mathcal{I}^{\text{syn}}(k)\Big) = \mathbb{P}\left(|\bar{x}_k - \mu| \leq \frac{\Phi^{-1}(1-\alpha/2)}{\sqrt{k}}\right)$$

$$= \mathbb{P}\left(|\sqrt{k}(\bar{x}_k - \mu^{\text{syn}}) - \sqrt{k}(\mu - \mu^{\text{syn}})| \leq \Phi^{-1}(1-\alpha/2)\right)$$

$$= \mathbb{P}\left(\sqrt{k}\Delta - \Phi^{-1}(1-\alpha/2) \leq \sqrt{k}(\bar{x}_k - \mu^{\text{syn}}) \leq \sqrt{k}\Delta + \Phi^{-1}(1-\alpha/2)\right).$$

Note that $\sqrt{k}(\bar{x}_k - \mu^{\text{syn}}) \sim N(0,1)$. Thus, whenever $\Delta \neq 0$,

$$\mathbb{P}\Big(\mu \in \mathcal{I}^{\text{syn}}(k)\Big) < 1 - \alpha,$$

failing to attain $(1-\alpha)$ coverage probability, regardless of the sample size $k$.

## B.2. Proof of Theorem 2.4

We will prove the following stronger guarantee.

**Lemma B.1** (Conditional coverage). *Consider the setting of Theorem 2.4. Let $\delta \in (0,1)$. With probability at least $1 - \delta$,*

$$\mathbb{P}\Big(\mu \in \mathcal{I}^{\text{syn}}(\widehat{k}) \,\Big|\, \widehat{k}\Big) \geq 1 - \alpha - \sqrt{\frac{2\log(1/\delta)}{m}}. \tag{19}$$

By Lemma B.1, we obtain

$$\mathbb{P}\Big(\mu \in \mathcal{I}^{\text{syn}}(\widehat{k})\Big) = \mathbb{E}\left[\mathbb{P}\Big(\mu \in \mathcal{I}^{\text{syn}}(\widehat{k}) \,\Big|\, \widehat{k}\Big)\right]$$

$$= \int_0^\infty \mathbb{P}\left(\mathbb{P}\Big(\mu \in \mathcal{I}^{\text{syn}}(\widehat{k}) \,\Big|\, \widehat{k}\Big) > t\right) dt$$

$$\geq \int_0^{1-\alpha} \left[1 - \exp\left(-\frac{m}{2}\left(t - (1-\alpha)\right)^2\right)\right] dt$$

$$\geq 1 - \alpha - \int_{-\infty}^{1-\alpha} \exp\left(-\frac{m}{2}\left(t - (1-\alpha)\right)^2\right) dt$$

$$\geq 1 - \alpha - \sqrt{\frac{2}{m}}.$$

We will now prove Lemma B.1. Define $\varepsilon = \sqrt{2\log(1/\delta)/m}$ and a deterministic oracle sample size

$$\bar{k} = \inf\left\{k \in [K] : \mathbb{P}\Big(\mu \notin \mathcal{I}^{\text{syn}}(k)\Big) > \alpha + \varepsilon\right\}. \tag{20}$$

If $\bar{k} = \inf \emptyset$ does not exist, then there is nothing to prove. Now suppose that $\bar{k} \in [K]$ exists. We will prove that with probability at least $1 - \delta$, it holds that $G(\bar{k}) > \alpha/2$. When this event happens, we have $\widehat{k} < \bar{k}$, which implies $\mathbb{P}\big(\mu \notin \mathcal{I}^{\text{syn}}(\widehat{k}) \mid \widehat{k}\big) \leq \alpha + \varepsilon$ and thus (19), thanks to the independence of $\widehat{k}$ and $(\psi, \mathcal{D}^{\text{syn}})$.

By Hoeffding's inequality (e.g., Theorem 2.8 in (Boucheron et al., 2013)) and the conditional independence of $(\mathcal{D}_1, \mathcal{D}_1^{\text{syn}}), ..., (\mathcal{D}_m, \mathcal{D}_m^{\text{syn}})$ given $(\psi_1, ..., \psi_m)$,

$$\mathbb{P}\left(G(\bar{k}) \geq \frac{1}{m}\sum_{j=1}^{m} \mathbb{P}\Big(\bar{y}_j \notin \mathcal{I}_j^{\text{syn}}(\bar{k})\Big) - \sqrt{\frac{\log(1/\delta)}{2m}}\right) \geq 1 - \delta. \tag{21}$$

We now bound $\mathbb{P}\big(\bar{y}_j \notin \mathcal{I}_j^{\mathsf{syn}}(\bar{k})\big)$. For each $j \in [m]$ and $k \in [K]$,

$$\mathbf{1}\left\{\bar{y}_j \notin \mathcal{I}_j^{\mathsf{syn}}(k)\right\} \geq \mathbf{1}\left\{\bar{y}_j < \mu_j \text{ and } \mu_j < \min \mathcal{I}_j^{\mathsf{syn}}(k)\right\} + \mathbf{1}\left\{\bar{y}_j \geq \mu_j \text{ and } \mu_j > \max \mathcal{I}_j^{\mathsf{syn}}(k)\right\}.$$

By the conditional independence of $\mathcal{D}_j$ and $\mathcal{D}_j^{\mathsf{syn}}$ given $\psi_j$,

$$\mathbb{P}\Big(\bar{y}_j < \mu_j \text{ and } \mu_j < \min \mathcal{I}_j^{\mathsf{syn}}(k)\Big)$$

$$= \mathbb{E}\Big[\mathbb{P}\Big(\bar{y}_j < \mu_j \,\Big|\, \psi_j\Big) \cdot \mathbb{P}\Big(\mu_j < \min \mathcal{I}_j^{\mathsf{syn}}(k) \,\Big|\, \psi_j\Big)\Big]$$

$$= \mathbb{E}\left[\frac{1}{2} \cdot \mathbb{P}\Big(\mu_j < \min \mathcal{I}_j^{\mathsf{syn}}(k) \,\Big|\, \psi_j\Big)\right]$$

$$= \frac{1}{2}\mathbb{P}\Big(\mu_j < \min \mathcal{I}_j^{\mathsf{syn}}(k)\Big).$$

Similarly,

$$\mathbb{P}\Big(\bar{y}_j \geq \mu_j \text{ and } \mu_j > \max \mathcal{I}_j^{\mathsf{syn}}(k)\Big) = \frac{1}{2}\mathbb{P}\Big(\mu_j > \max \mathcal{I}_j^{\mathsf{syn}}(k)\Big).$$

Therefore,

$$\mathbb{P}\Big(\bar{y}_j \notin \mathcal{I}_j^{\mathsf{syn}}(k)\Big) \geq \frac{1}{2}\left[\mathbb{P}\Big(\mu_j < \min \mathcal{I}_j^{\mathsf{syn}}(k)\Big) + \mathbb{P}\Big(\mu_j > \max \mathcal{I}_j^{\mathsf{syn}}(k)\Big)\right]$$

$$= \frac{1}{2}\mathbb{P}\Big(\mu_j \notin \mathcal{I}_j^{\mathsf{syn}}(k)\Big) = \frac{1}{2}\mathbb{P}\Big(\mu \notin \mathcal{I}^{\mathsf{syn}}(k)\Big). \tag{22}$$

When the event in (21) happens,

$$G(\bar{k}) \geq \frac{1}{m} \sum_{j=1}^{m} \mathbb{P}\Big(\bar{y}_j \notin \mathcal{I}_j^{\mathsf{syn}}(\bar{k})\Big) - \sqrt{\frac{\log(1/\delta)}{2m}}$$

$$\geq \frac{1}{2}\mathbb{P}\Big(\mu \notin \mathcal{I}^{\mathsf{syn}}(\bar{k})\Big) - \sqrt{\frac{\log(1/\delta)}{2m}} \qquad \text{(by (22))}$$

$$> \frac{\alpha}{2}. \qquad \text{(by definition of } \bar{k}\text{)}$$

This completes the proof.

### B.3. Proof of Theorem 2.6

By Hoeffding's inequality (e.g., Theorem 2.8 in (Boucheron et al., 2013)) and a union bound, the following happens with probability at least $1 - \delta$:

$$G(k) \leq \frac{1}{m} \sum_{j=1}^{m} \mathbb{P}\Big(\bar{y}_j \notin \mathcal{I}_j^{\mathsf{syn}}(k)\Big) + \sqrt{\frac{\log(n/\delta)}{2m}}, \quad \forall k \leq \min\left\{n, K, \left(\frac{C}{5\Delta}\right)^2\right\} \tag{23}$$

We now show that the right hand side of (23) is at most $\alpha/2$ for $m$ large. For all $j \in [m]$ and $k \in [K]$,

$$\mathbb{P}\Big(\bar{y}_j \notin \mathcal{I}_j^{\mathsf{syn}}(k)\Big) = \mathbb{P}\left(|\bar{y}_j - \bar{y}_{j,k}^{\mathsf{syn}}| > \frac{C}{\sqrt{k}}\right).$$

Since $\bar{y}_j \sim N(\mu_j, 1/n)$ and $\bar{y}_{j,k}^{\mathsf{syn}} \sim N(\mu_j^{\mathsf{syn}}, 1/k)$, then

$$\bar{y}_j - \bar{y}_{j,k}^{\mathsf{syn}} \sim N\left(\mu_j - \mu_j^{\mathsf{syn}}, \frac{1}{k} + \frac{1}{n}\right).$$

When $k \leq \min\{n, K, (\frac{C}{5\Delta})^2\}$, we have

$$\mathbb{P}\Big(\bar{y}_j \notin \mathcal{I}_j^{\mathsf{syn}}(k)\Big) \leq \mathbb{P}\left(|(\bar{y}_j - \bar{y}_{j,k}^{\mathsf{syn}}) - (\mu_j - \mu_j^{\mathsf{syn}})| + \Delta > \frac{C}{\sqrt{k}}\right)$$

$$= 2\Phi\left(-\frac{C/\sqrt{k} - \Delta}{\sqrt{k^{-1} + n^{-1}}}\right) \leq 2\Phi\left(-\frac{4C/(5\sqrt{k})}{\sqrt{k^{-1} + k^{-1}}}\right) \qquad (\Delta \leq C/(5\sqrt{k}))$$

$$= 2\Phi\left(-\frac{2\sqrt{2}C}{5}\right) = 2\Phi\left(-\frac{4\sqrt{2}\Phi^{-1}(1 - \alpha/4)}{5}\right) = 2\Phi\left(\frac{4\sqrt{2}\Phi^{-1}(\alpha/4)}{5}\right) < \frac{\alpha}{2}.$$

Let

$$\xi = \frac{\alpha}{2} - 2\Phi\left(\frac{4\sqrt{2}\Phi^{-1}(\alpha/4)}{5}\right),$$

then $\xi > 0$ and

$$\mathbb{P}\Big(\bar{y}_j \notin \mathcal{I}_j^{\mathsf{syn}}(k)\Big) \leq \frac{\alpha}{2} - \xi. \qquad (24)$$

When $m > \log(n/\delta)/(2\xi^2)$, substituting (24) into (23) yields that for all $k \leq \min\{n, K, (\frac{C}{5\Delta})^2\}$,

$$G(k) \leq \frac{1}{m}\sum_{j=1}^{m}\mathbb{P}\Big(\bar{y}_j \notin \mathcal{I}_j^{\mathsf{syn}}(k)\Big) + \sqrt{\frac{\log(n/\delta)}{2m}}$$

$$< \left(\frac{\alpha}{2} - \xi\right) + \xi = \frac{\alpha}{2}.$$

When this happens, we have $\widehat{k} \geq \min\{n, K, (\frac{C}{5\Delta})^2\}$.

## B.4. Proof of Theorem 3.3

We will prove the following stronger guarantee.

**Lemma B.2** (Conditional coverage). *Consider the setting of Theorem 3.3. Let $\delta \in (0, 1)$. With probability at least $1 - \delta$,*

$$\mathbb{P}\Big(\theta(\psi) \in \mathcal{S}^{\mathsf{syn}}(\widehat{k}) \,\Big|\, \widehat{k}\Big) \geq 1 - \alpha - \gamma^{-1}\sqrt{\frac{\log(1/\delta)}{2m}}. \qquad (25)$$

By Lemma B.1, we obtain

$$\mathbb{P}\Big(\theta(\psi) \in \mathcal{S}^{\mathsf{syn}}(\widehat{k})\Big) = \mathbb{E}\left[\mathbb{P}\Big(\theta(\psi) \in \mathcal{S}^{\mathsf{syn}}(\widehat{k}) \,\Big|\, \widehat{k}\Big)\right]$$

$$= \int_0^\infty \mathbb{P}\Big(\mathbb{P}\Big(\theta(\psi) \in \mathcal{S}^{\mathsf{syn}}(\widehat{k}) \,\Big|\, \widehat{k}\Big) > t\Big)\, dt$$

$$\geq \int_0^{1-\alpha} \left[1 - \exp\left(-2m\gamma^2\big(t - (1-\alpha)\big)^2\right)\right]\, dt$$

$$\geq 1 - \alpha - \int_{-\infty}^{1-\alpha} \exp\left(-2m\gamma^2\big(t - (1-\alpha)\big)^2\right)\, dt$$

$$= 1 - \alpha - \sqrt{\frac{\pi}{8}} \cdot \gamma^{-1}\sqrt{\frac{1}{m}}$$

$$\geq 1 - \alpha - \gamma^{-1}\sqrt{\frac{1}{2m}}.$$

We now prove Lemma B.2. Define $\varepsilon = \gamma^{-1}\sqrt{\frac{\log(1/\delta)}{2m}}$ and a deterministic oracle sample size

$$\bar{k} = \inf\left\{k \in [K] : \mathbb{P}\Big(\theta(\psi) \notin \mathcal{S}^{\mathsf{syn}}(k)\Big) > \alpha + \varepsilon\right\}. \qquad (26)$$

If $\bar{k} = \inf \emptyset$ does not exist, then there is nothing to prove. Now suppose $\bar{k} \in [K]$ exists. We will prove that with probability at least $1 - \delta$, it holds that $L(\bar{k}) > \gamma\alpha$. When this event happens, we have $\widehat{k} < \bar{k}$, which implies $\mathbb{P}\big(\theta(\psi) \notin \mathcal{S}^{\mathsf{syn}}(\widehat{k}) \mid \widehat{k}\big) \leq \alpha + \varepsilon$ and thus (25), thanks to the independence of $\widehat{k}$ and $(\psi, \mathcal{D}^{\mathsf{syn}})$.

By Hoeffding's inequality (e.g., Theorem 2.8 in (Boucheron et al., 2013)) and the conditional independence of $(\mathcal{D}_1, \mathcal{D}_1^{\mathsf{syn}}), ..., (\mathcal{D}_m, \mathcal{D}_m^{\mathsf{syn}})$ given $(\psi_1, ..., \psi_m)$,

$$\mathbb{P}\left( L(\bar{k}) \geq \frac{1}{m} \sum_{j=1}^{m} \mathbb{P}\Big(\mathcal{S}_j \nsubseteq \mathcal{S}_j^{\mathsf{syn}}(\bar{k})\Big) - \sqrt{\frac{\log(1/\delta)}{2m}} \right) \geq 1 - \delta. \tag{27}$$

We now bound $\mathbb{P}\big(\mathcal{S}_j \nsubseteq \mathcal{S}_j^{\mathsf{syn}}(\bar{k})\big)$. For each $j \in [m]$ and $j \in [K]$,

$$\mathbf{1}\left\{ \mathcal{S}_j \nsubseteq \mathcal{S}_j^{\mathsf{syn}}(k) \right\} \geq \mathbf{1}\left\{ \theta(\psi_j) \in \mathcal{S}_j \text{ and } \theta(\psi_j) \notin \mathcal{S}_j^{\mathsf{syn}}(k) \right\}.$$

By the conditional independence of $\mathcal{D}_j$ and $\mathcal{D}_j^{\mathsf{syn}}$ given $\psi_j$,

$$\begin{aligned}
\mathbb{P}\Big(\mathcal{S}_j \nsubseteq \mathcal{S}_j^{\mathsf{syn}}(k)\Big) &\geq \mathbb{E}\Big[ \mathbb{P}\Big( \theta(\psi_j) \in \mathcal{S}_j \text{ and } \theta(\psi_j) \notin \mathcal{S}_j^{\mathsf{syn}}(k) \mid \psi_j \Big) \Big] \\
&= \mathbb{E}\Big[ \mathbb{P}\Big( \theta(\psi_j) \in \mathcal{S}_j \mid \psi_j \Big) \cdot \mathbb{P}\Big( \theta(\psi_j) \notin \mathcal{S}_j^{\mathsf{syn}}(k) \mid \psi_j \Big) \Big] \\
&\geq \mathbb{E}\Big[ \gamma \cdot \mathbb{P}\Big( \theta(\psi_j) \notin \mathcal{S}_j^{\mathsf{syn}}(k) \mid \psi_j \Big) \Big] \\
&= \gamma \cdot \mathbb{P}\Big( \theta(\psi_j) \notin \mathcal{S}_j^{\mathsf{syn}}(k) \Big) \\
&= \gamma \cdot \mathbb{P}\Big( \theta(\psi) \notin \mathcal{S}^{\mathsf{syn}}(k) \Big). 
\end{aligned} \tag{28}$$

Therefore, when the event in (27) happens,

$$\begin{aligned}
L(\bar{k}) &\geq \frac{1}{m} \sum_{j=1}^{m} \mathbb{P}\Big(\mathcal{S}_j \nsubseteq \mathcal{S}_j^{\mathsf{syn}}(\bar{k})\Big) - \sqrt{\frac{\log(1/\delta)}{2m}} \\
&\geq \gamma \cdot \mathbb{P}\Big( \theta(\psi) \notin \mathcal{S}^{\mathsf{syn}}(\bar{k}) \Big) - \sqrt{\frac{\log(1/\delta)}{2m}} && \text{(by (28))} \\
&> \gamma\alpha. && \text{(by definition of } \bar{k}\text{)}
\end{aligned}$$

This completes the proof.

## C. Details of Numerical Experiments

### C.1. The OpinionQA Dataset

**Selection of survey questions.** The original dataset is categorized into topics such as health, crime/security, and political issues. Ideally, we would want to consider questions from the same category to ensure that they are similar enough. However, the category with the most questions has fewer than 200 questions. We thus consider pooling all questions. The dataset has 1,442 survey questions in total, which is too large for our computational resources. We selected a subset of questions as follows. First, while the number of choices ranges from 2 to 19, most questions have 5 choices. To give a fair comparison and for simplicity, we only consider questions with 5 choices. Second, not all questions have choices that can be clearly ordered in sentiments, such as the following one:

*Who do you think has the most responsibility to reduce the amount of made-up news and information? 1. The government, 2. Technology companies, 3. The public, 4. The news media, 5. None of these, 6. Refused.*

We asked GPT-4o to determine if a question's choices can be ordered in sentiments and we keep those that have GPT-4o's affirmative answer. This leaves us with 546 questions. To compensate for the loss of similarity by pooling questions across

various topics and to further reduce our computational cost, we selected 400 questions that are "most similar" to each other by embedding the question statements using OpenAI's `text-embedding-3-small`, calculating the mean, and selecting the 400 questions with the smallest Euclidean distance to the mean. Out of these 400 questions, 15 questions have various issues with their choices by manual inspection, so we exclude them. This leaves us with 385 questions. All these questions happen to have at least 400 responses.

**Example questions.** The questions in the OpinionQA dataset span a wide range of topics, including health, crime/security, and political issues. Some example questions are as follows:

- *How much, if at all, do you think wages and incomes are contributing to your opinion about how the economy is doing?*

  *1. A great deal    2. A fair amount    3. Not too much    4. Not at all    5. Refused*

- *Regardless of whether you would want to move, how likely is it that you will move to a different community at some point in the future?*

  *1. Very likely    2. Somewhat likely    3. Not too likely    4. Not at all likely    5. Refused*

- *How much, if anything, would you be willing to change about how you live and work to help reduce the effects of global climate change? Would you be willing to make:*

  *1. A lot of changes  2. Some changes  3. Only a few changes  4. No changes at all  5. Refused*

**Profiles.** Excluding surveyees with missing information, each of the 385 questions we consider has at least 400 responses. Since there was no information on the surveyees' identification, by dropping repeated profiles we can only say that there are at least 32,864 surveyees. Each surveyee is described by 12 features. Their corresponding categories are listed in Table 1.

*Table 1.* Categories of surveyees' features in the OpinionQA dataset.

| Feature | Options |
|---|---|
| US citizenship | 'Yes', 'No' |
| Region | 'Northeast', 'Midwest', 'South', 'West' |
| Sex | 'Male', 'Female' |
| Age | '18-29', '30-49', '50-64', '65+' |
| Marital status | 'Married', 'Divorced', 'Separated', 'Widowed', 'Never been married' |
| Race | 'White', 'Black', 'Asian', 'Hispanic', 'Other' |
| Educational background | 'Less than high school', 'High school graduate', 'Some college, no degree', 'Associate's degree', 'College graduate/some postgrad', 'Postgraduate' |
| Income | 'Less than $30,000', '$30,000-$50,000', '$50,000-$75,000', '$75,000-$100,000', '$100,000 or more' |
| Religious affiliation | 'Protestant', 'Roman Catholic', 'Mormon', 'Orthodox', 'Jewish', 'Muslim', 'Buddhist', 'Hindu', 'Atheist', 'Agnostic', 'Other', 'Nothing in particular' |
| Religious attendance | 'More than once a week', 'Once a week', 'Once or twice a month', 'A few times a year', 'Seldom', 'Never' |
| Political party | 'Republican', 'Democrat', 'Independent', 'Other' |
| Political ideology | 'Very conservative', 'Conservative', 'Moderate', 'Liberal', 'Very liberal' |

**Synthetic response generation.** We generate synthetic profiles by bootstrapping the 32,864 unique real profiles. We then generate synthetic answers by prompting LLMs to pretend that they are a surveyee with the synthetic profile and answer the question. An example prompt is as follows:

> *Pretend that you reside in the US and you are a US citizen from the West region of the country. You are female, your age is between 18 and 29, and you are single. In terms of race, you are white. In terms of education, you attended college but did not graduate. Your annual income is less than $30,000. Religion-wise, you do not belong to any particular religion, and you never attend religious services. Politically, you are affiliated with a political party that is not Democratic or Republican, and you consider your political ideology to be liberal. Please answer the following question:*
>
> *How much, if at all, do you think what happens to black people in the country overall affects what happens in your own life? ['1. A lot', '2. Some', '3. Not much', '4. Not at all', '5. Refused'].*
>
> *Please provide your answer choice (a single number from 1 to 5) in double square brackets.*

In our experiments, LLMs usually directly gave answers in the required format, e.g., '[[2]]'.

### C.2. The EEDI Dataset

**Example questions.** Some example questions from the EEDI dataset are as follows:

- *What number belongs in the box?* $\Box + 7 = 2$

  $$A)\ 9 \quad B)\ \text{-}5 \quad C)\ \text{-}6 \quad D)\ 5$$

- *If you multiply a square number by* 9*, you get a square number. Is this statement:*

  $$A)\ \text{always true} \quad B)\ \text{sometimes true} \quad C)\ \text{never true} \quad D)\ \text{impossible to say}$$

- *Which calculation is equal to* $-20$*?*

  $$A)\ 2 \times (-2) - (-4) \times 4 \quad B)\ -28 - (-4) \times 2 \quad C)\ (-5)\top2 + 5 \quad D)\ (-42) \div (-2) + 1$$

**Profile distribution.** Excluding students with missing information which take up less than 10% of the total population, there are 2,111 students who answered at least one of the 412 questions. Each student is described by three features: gender, age, and whether or not they are eligible for free school meals or premium pupil. Gender is represented by 1 or 2, where 1 corresponds to female and 2 corresponds to male. The students' ages are rounded to integers from 11 and 14. Whether or not a student is eligible for free school meals is represented by 0 or 1, where 0 corresponds to not eligible and 1 corresponds to eligible. The distribution of these students' features is presented in Table 2.

*Table 2.* Summary statistics of students' features in the EEDI dataset.

|  | min | max | mean | median | standard deviation |
|---|---|---|---|---|---|
| Gender | 1 | 2 | 1.4988 | 1 | 0.5001 |
| Age | 11 | 14 | 11.2776 | 11 | 0.4696 |
| Premium Pupil | 0 | 1 | 0.2842 | 0 | 0.4512 |

**Synthetic response generation.** For each question, we generate synthetic profiles by sampling with replacement from the real profiles. We then generate synthetic answers by prompting LLMs to pretend that they are a student with the synthetic profile and answer the question. We adapted the prompt from (He-Yueya et al., 2024) with slight modifications to reduce computational cost. An example prompt featuring an 11-year-old boy who is not eligible for free school meals is as follows:

> *Pretend that you are an 11-year-old student. Your gender is male. You are not eligible for free school meals or pupil premium due to being relatively financially advantaged. Given your characteristics, is it likely that you would be able to solve the following problem?*

*Problem: [Insert question here]*

*If yes, put the final answer choice (a single letter) in double square brackets. If you are likely to struggle with this problem, put a plausible incorrect answer choice (a single letter) in double square brackets.*

An example answer from GTP-4o when given the second example question above is as follows:

*As an 11-year-old student, I might have learned about square numbers and multiplication in school. However, the problem may be a bit tricky if I haven't thought about how multiplying square numbers by other numbers can also result in square numbers. I might not immediately realize that 9 is actually a square number itself (3 squared), which makes this property more evident.*

*Considering this, I could find the reasoning challenging and decide based on a misconception. I might go with a plausible incorrect answer choice like [[B]] because I might think that it's only sometimes possible without realizing the full mathematical principle involved.*

### C.3. Visualization of Simulation Sample Size Selection

In Figure 5, we visualize the process of selecting the simulation sample size $\widehat{k}$: it is the first $k$ at which $\{L(k)\}_{k=1}^{K}$ up-crosses the threshold $\gamma\alpha$.

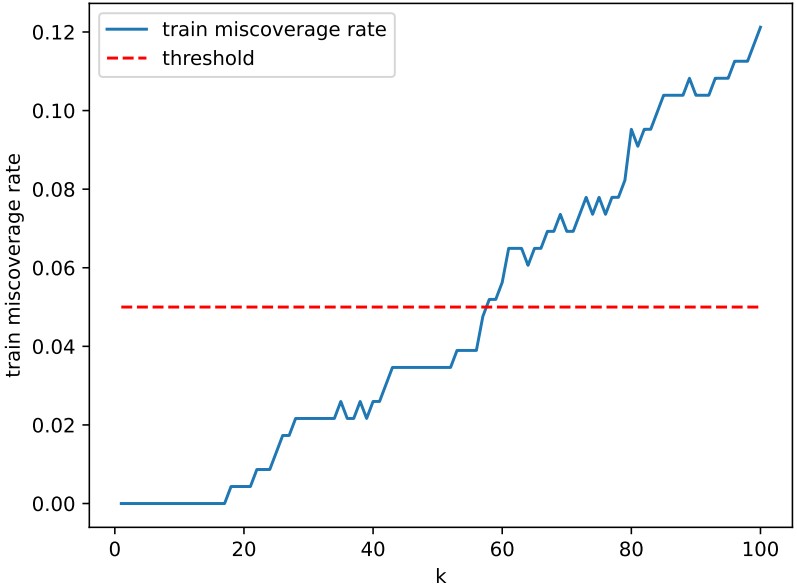

*Figure 5.* Visualization of simulation sample size selection for the OpinionQA dataset. Here $\alpha = 0.1$, $\gamma = 0.5$, $K = 100$, $c = \sqrt{2}$ and $M = 2$. The red dotted horizontal line is threshold $\gamma\alpha = 0.05$. The blue curve represents $\{L(k)\}_{k=1}^{K}$. Our approach chooses $\widehat{k}$ as the point at which the blue line first up-crosses the red threshold: $\widehat{k} = 57$.

## D. Additional Experiment Results

In this section, we provide additional experiment results for three performance metrics: the miscoverage probability proxy $\widetilde{L}(\widehat{k})$ as defined in (18), the selected simulation sample size $\widehat{k}$, and the half-width of the synthetic confidence interval $\mathcal{S}^{\mathsf{syn}}(\widehat{k})$.

## D.1. Miscoverage Probability Proxy

In Table 3 and Table 4, we present the $p$-values for the hypothesis test $\mathbb{E}\big[\widetilde{L}(\widehat{k})\big] \leq \alpha$ against $\mathbb{E}\big[\widetilde{L}(\widehat{k})\big] > \alpha$ for various LLMs and $\alpha$ over the OpinionQA and EEDI datasets, respectively. In Table 5 and Table 6, we present the means and standard errors of $\widetilde{L}(\widehat{k})$ as defined by (18) over 100 random splits. They complement Figure 2.

*Table 3.* $p$-values for the hypothesis test $\mathbb{E}\big[\widetilde{L}(\widehat{k})\big] \leq \alpha$ against $\mathbb{E}\big[\widetilde{L}(\widehat{k})\big] > \alpha$ for various LLMs and $\alpha$ over the OpinionQA dataset. Note that the $\alpha$ values specify the experiment configuration and are not the significance levels of the hypothesis test.

| $\alpha$ | 0.05 | 0.10 | 0.15 | 0.20 | 0.25 | 0.30 | 0.35 | 0.40 | 0.45 | 0.50 |
|---|---|---|---|---|---|---|---|---|---|---|
| GPT-3.5-turbo | 0.0618 | 0.1293 | 0.1996 | 0.2245 | 0.6981 | 0.3460 | 0.3387 | 0.2591 | 0.7846 | 0.8430 |
| GPT-4o-mini | 0.1619 | 0.1642 | 0.0658 | 0.2352 | 0.4741 | 0.4363 | 0.2000 | 0.1150 | 0.5717 | 0.3812 |
| GPT-4o | 0.3679 | 0.3218 | 0.4910 | 0.2243 | 0.5837 | 0.4379 | 0.2329 | 0.3388 | 0.9686 | 0.9226 |
| Claude 3.5 Haiku | 0.3857 | 0.5889 | 0.5945 | 0.4558 | 0.7030 | 0.6029 | 0.7370 | 0.8853 | 0.9922 | 0.9791 |
| Llama-3-8B | 0.7571 | 0.4376 | 0.2609 | 0.6072 | 0.5689 | 0.4178 | 0.7018 | 0.6132 | 0.7391 | 0.5169 |
| Llama 3.3 70B | 0.5505 | 0.6519 | 0.7687 | 0.5936 | 0.7055 | 0.7207 | 0.6050 | 0.3530 | 0.6973 | 0.5274 |
| Mistral 7B | 0.6249 | 0.0505 | 0.0517 | 0.0333 | 0.6473 | 0.5988 | 0.5394 | 0.2119 | 0.3380 | 0.3426 |
| DeepSeek-V3 | 0.0686 | 0.0969 | 0.0841 | 0.1513 | 0.5075 | 0.2257 | 0.1678 | 0.2638 | 0.3457 | 0.3024 |
| Random | 0.7779 | 0.6214 | 0.7527 | 0.6586 | 0.9570 | 0.9903 | 0.9321 | 0.7984 | 0.9259 | 0.9783 |

*Table 4.* $p$-values for the hypothesis test $\mathbb{E}\big[\widetilde{L}(\widehat{k})\big] \leq \alpha$ against $\mathbb{E}\big[\widetilde{L}(\widehat{k})\big] > \alpha$ for various LLMs and $\alpha$ over the EEDI dataset. Note that the $\alpha$ values specify the experiment configuration and are not the significance levels of the hypothesis test.

| $\alpha$ | 0.05 | 0.10 | 0.15 | 0.20 | 0.25 | 0.30 | 0.35 | 0.40 | 0.45 | 0.50 |
|---|---|---|---|---|---|---|---|---|---|---|
| GPT-3.5-turbo | 0.9988 | 1.0000 | 1.0000 | 1.0000 | 1.0000 | 1.0000 | 1.0000 | 1.0000 | 1.0000 | 1.0000 |
| GPT-4o-mini | 1.0000 | 1.0000 | 1.0000 | 1.0000 | 1.0000 | 1.0000 | 1.0000 | 1.0000 | 1.0000 | 1.0000 |
| GPT-4o | 0.1832 | 0.5820 | 0.9998 | 0.9732 | 0.9938 | 0.9807 | 0.9292 | 0.8259 | 0.9757 | 0.9922 |
| Claude 3.5 Haiku | 0.8656 | 0.9999 | 1.0000 | 1.0000 | 1.0000 | 1.0000 | 1.0000 | 1.0000 | 1.0000 | 1.0000 |
| Llama 3.3 70B | 0.9575 | 1.0000 | 1.0000 | 1.0000 | 1.0000 | 1.0000 | 1.0000 | 1.0000 | 1.0000 | 1.0000 |
| Mistral 7B | 1.0000 | 1.0000 | 1.0000 | 1.0000 | 1.0000 | 1.0000 | 1.0000 | 1.0000 | 1.0000 | 1.0000 |
| DeepSeek-V3 | 0.9444 | 0.9789 | 0.9789 | 0.9981 | 1.0000 | 0.9992 | 1.0000 | 1.0000 | 1.0000 | 1.0000 |
| Random | 0.9800 | 1.0000 | 1.0000 | 0.9997 | 0.9954 | 0.9895 | 0.9979 | 0.9997 | 0.9920 | 0.9990 |

*Table 5.* Means and standard errors of $\widetilde{L}(\widehat{k})$ over 100 random splits for various LLMs and $\alpha$ over the OpinionQA dataset.

| $\alpha$ | 0.05 | 0.10 | 0.15 | 0.20 | 0.25 | 0.30 | 0.35 | 0.40 | 0.45 | 0.50 |
|---|---|---|---|---|---|---|---|---|---|---|
| GPT-3.5 -turbo | 0.0547 (0.003) | 0.1051 (0.0045) | 0.1549 (0.0059) | 0.2051 (0.0067) | 0.2461 (0.0075) | 0.3032 (0.0082) | 0.3536 (0.0087) | 0.4056 (0.0086) | 0.4432 (0.0086) | 0.4909 (0.009) |
| GPT-4o -mini | 0.0535 (0.0036) | 0.1052 (0.0053) | 0.1588 (0.0059) | 0.2052 (0.0072) | 0.2505 (0.008) | 0.3014 (0.0089) | 0.3574 (0.0088) | 0.4104 (0.0087) | 0.4484 (0.0086) | 0.5027 (0.009) |
| GPT-4o | 0.051 (0.0031) | 0.1022 (0.0048) | 0.1501 (0.0058) | 0.2049 (0.0065) | 0.2486 (0.0068) | 0.3012 (0.0075) | 0.3555 (0.0075) | 0.4031 (0.0075) | 0.4361 (0.0075) | 0.489 (0.0078) |
| Claude-3-5 -Haiku | 0.051 (0.0036) | 0.0988 (0.0052) | 0.1484 (0.0065) | 0.2008 (0.007) | 0.246 (0.0076) | 0.2979 (0.008) | 0.3448 (0.0082) | 0.3899 (0.0084) | 0.4284 (0.0089) | 0.4813 (0.0092) |
| Llama-3-8B | 0.0481 (0.0028) | 0.1006 (0.0041) | 0.1536 (0.0057) | 0.1982 (0.0067) | 0.2487 (0.0075) | 0.3017 (0.0081) | 0.3452 (0.0091) | 0.3974 (0.009) | 0.4442 (0.0091) | 0.4996 (0.0092) |
| Llama-3.3 -70B | 0.0496 (0.0031) | 0.0983 (0.0043) | 0.1458 (0.0057) | 0.1984 (0.0066) | 0.2464 (0.0067) | 0.2955 (0.0078) | 0.3479 (0.0078) | 0.4032 (0.0086) | 0.4453 (0.009) | 0.4994 (0.0094) |
| Mistral-7B | 0.049 (0.0033) | 0.1075 (0.0046) | 0.1586 (0.0053) | 0.21 (0.0055) | 0.2478 (0.0058) | 0.2983 (0.0067) | 0.3492 (0.0079) | 0.4065 (0.0081) | 0.4536 (0.0087) | 0.5035 (0.0087) |
| DeepSeek-V3 | 0.0549 (0.0033) | 0.106 (0.0046) | 0.1571 (0.0052) | 0.2062 (0.006) | 0.2499 (0.0069) | 0.3051 (0.0067) | 0.3579 (0.0082) | 0.4053 (0.0084) | 0.4535 (0.0088) | 0.5049 (0.0095) |
| Random | 0.0478 (0.0029) | 0.0986 (0.0046) | 0.1462 (0.0055) | 0.1975 (0.006) | 0.2403 (0.0057) | 0.2844 (0.0067) | 0.3412 (0.0059) | 0.3948 (0.0062) | 0.4401 (0.0068) | 0.4853 (0.0073) |

*Table 6.* Means and standard errors of $\widetilde{L}(\widehat{k})$ over 100 random splits for various LLMs and $\alpha$ over the EEDI dataset.

| $\alpha$ | 0.05 | 0.10 | 0.15 | 0.20 | 0.25 | 0.30 | 0.35 | 0.40 | 0.45 | 0.50 |
|---|---|---|---|---|---|---|---|---|---|---|
| GPT-3.5 -turbo | 0.0401 (0.0033) | 0.0882 (0.0025) | 0.1122 (0.0057) | 0.1401 (0.0044) | 0.2067 (0.0038) | 0.26 (0.0051) | 0.308 (0.0088) | 0.2806 (0.0114) | 0.3655 (0.0116) | 0.3925 (0.0104) |
| GPT-4o -mini | 0.035 (0.0033) | 0.0773 (0.0026) | 0.1155 (0.0066) | 0.1463 (0.006) | 0.2074 (0.0055) | 0.2425 (0.0037) | 0.2909 (0.0043) | 0.3395 (0.0108) | 0.3417 (0.0117) | 0.3747 (0.0109) |
| GPT-4o | 0.0528 (0.0032) | 0.099 (0.0047) | 0.1365 (0.0039) | 0.1891 (0.0057) | 0.2313 (0.0075) | 0.283 (0.0082) | 0.3383 (0.008) | 0.3931 (0.0074) | 0.4342 (0.008) | 0.4805 (0.0081) |
| Claude-3-5 -Haiku | 0.0461 (0.0036) | 0.0833 (0.0044) | 0.1234 (0.0046) | 0.1618 (0.0055) | 0.1993 (0.0072) | 0.2577 (0.008) | 0.2772 (0.008) | 0.3326 (0.0081) | 0.3948 (0.0089) | 0.4327 (0.0081) |
| Llama-3.3 -70B | 0.0444 (0.0033) | 0.0785 (0.0031) | 0.1261 (0.0055) | 0.1355 (0.0059) | 0.2096 (0.0054) | 0.2564 (0.0051) | 0.2998 (0.0059) | 0.348 (0.0061) | 0.3988 (0.0085) | 0.4235 (0.0133) |
| Mistral-7B | 0.0366 (0.0033) | 0.0829 (0.0043) | 0.1064 (0.003) | 0.1621 (0.0031) | 0.2074 (0.008) | 0.2447 (0.0098) | 0.2716 (0.0108) | 0.2858 (0.0055) | 0.3337 (0.0043) | 0.391 (0.005) |
| DeepSeek-V3 | 0.045 (0.0032) | 0.0913 (0.0043) | 0.1399 (0.005) | 0.1835 (0.0057) | 0.2153 (0.0068) | 0.2777 (0.0071) | 0.3223 (0.0067) | 0.3667 (0.0069) | 0.4166 (0.0075) | 0.46 (0.0073) |
| Random | 0.0446 (0.0026) | 0.0645 (0.0038) | 0.1235 (0.0064) | 0.1812 (0.0055) | 0.2338 (0.0062) | 0.2851 (0.0065) | 0.3299 (0.007) | 0.3702 (0.0086) | 0.4302 (0.0082) | 0.4731 (0.0087) |

## D.2. Selected Simulation Sample Size

In Table 7 and Table 8, we present the detailed results for the selected simulation sample size $\widehat{k}$. They complement Figure 3.

*Table 7.* Average $\widehat{k}$ (with 95% margin of error in parentheses) for various LLMs and various $\alpha$ over the OpinionQA dataset. GPT-4o has the largest $\widehat{k}$ on average.

| $\alpha$ | 0.05 | 0.10 | 0.15 | 0.20 | 0.25 | 0.30 | 0.35 | 0.40 | 0.45 | 0.50 |
|---|---|---|---|---|---|---|---|---|---|---|
| GPT-3.5 -turbo | 36.65 (1.064) | 44.03 (0.9373) | 44.62 (0.4879) | 44.22 (0.4263) | 43.59 (0.4794) | 43.21 (0.5823) | 43.44 (0.7403) | 44.75 (0.7367) | 45.22 (0.6492) | 45.91 (0.5296) |
| GPT-4o -mini | 26.85 (0.7517) | 33.68 (0.8189) | 37.37 (0.5871) | 37.56 (0.4058) | 37.08 (0.5025) | 38.4 (0.6345) | 40.09 (0.6149) | 41.36 (0.5745) | 41.6 (0.4671) | 41.51 (0.5328) |
| GPT-4o | **54.1** (**1.7862**) | **62.19** (**1.2286**) | **65.35** (**1.0516**) | **68.77** (**0.9885**) | **69.58** (**1.1095**) | **74.29** (**1.5015**) | **81.83** (**1.6431**) | **88.7** (**1.5013**) | **89.37** (**1.2831**) | **91.86** (**1.2268**) |
| Claude-3-5 -Haiku | 35.75 (1.0428) | 40.7 (0.7842) | 42.46 (0.7336) | 44.74 (0.6872) | 45.07 (0.6561) | 46.27 (0.6464) | 47.79 (0.7881) | 49.84 (0.8488) | 50.37 (0.7415) | 51.34 (0.6819) |
| Llama-3-8B | 28.81 (0.6264) | 31.93 (0.7149) | 34.35 (0.4828) | 34.83 (0.5382) | 35.13 (0.6855) | 37.3 (0.5248) | 38.1 (0.5491) | 38.9 (0.6747) | 40.5 (0.8084) | 44.25 (0.8016) |
| Llama-3.3 -70B | 46.3 (0.6921) | 47.67 (0.5339) | 46.39 (0.5331) | 48.12 (1.0276) | 53.49 (1.2524) | 56.21 (0.8797) | 57.45 (0.7294) | 58.04 (0.7947) | 58.62 (1.0285) | 61.8 (1.0627) |
| Mistral-7B | 30.81 (0.2566) | 33.47 (0.8928) | 39.47 (0.8815) | 44.59 (0.8935) | 47.53 (0.8622) | 52.43 (1.0825) | 56.34 (1.0113) | 58.43 (0.9535) | 60.29 (1.1178) | 64.85 (1.2481) |
| DeepSeek-V3 | 37.59 (1.4767) | 46.75 (0.79) | 46.66 (0.4771) | 45.1 (0.6201) | 46.03 (0.5657) | 47.77 (0.6135) | 47.76 (0.4606) | 47.61 (0.48) | 47.92 (0.6989) | 49.51 (0.7031) |
| Random | 17.09 (0.3761) | 18.07 (0.4402) | 19.33 (0.3361) | 19.83 (0.4046) | 19.85 (0.3261) | 20.36 (0.3227) | 20.01 (0.3191) | 19.92 (0.2849) | 19.93 (0.2662) | 20.4 (0.2252) |

*Table 8.* Average $\widehat{k}$ (with 95% margin of error in parentheses) for various LLMs and various $\alpha$ over the EEDI dataset. DeepSeek-V3 has the largest $\widehat{k}$ on average.

| $\alpha$ | 0.05 | 0.10 | 0.15 | 0.20 | 0.25 | 0.30 | 0.35 | 0.40 | 0.45 | 0.50 |
|---|---|---|---|---|---|---|---|---|---|---|
| GPT-3.5 -turbo | 5.68 (0.1033) | 5.89 (0.0613) | 5.19 (0.0769) | 5.03 (0.0334) | 4.99 (0.0195) | 4.94 (0.0465) | 4.71 (0.0889) | 4.17 (0.0736) | 4.29 (0.0889) | 4.13 (0.0659) |
| GPT-4o -mini | 6.26 (0.086) | 5.94 (0.0465) | 5.44 (0.0973) | 5.14 (0.068) | 5.06 (0.0465) | 5.0 (0) | 4.99 (0.0195) | 4.6 (0.096) | 4.25 (0.0849) | 4.15 (0.07) |
| GPT-4o | 8.04 (0.1438) | 7.82 (0.1672) | 8.31 (0.1853) | 8.1 (0.116) | 7.81 (0.1609) | 7.83 (0.152) | 7.84 (0.1459) | 7.93 (0.1308) | 7.9 (0.1224) | 8.17 (0.1178) |
| Claude-3-5 -Haiku | 7.57 (0.1337) | 7.42 (0.0967) | 7.05 (0.0802) | 6.82 (0.0849) | 6.59 (0.0964) | 6.47 (0.1017) | 6.15 (0.07) | 6.15 (0.07) | 6.12 (0.0889) | 6.04 (0.0674) |
| Llama-3.3 -70B | 6.19 (0.0863) | 5.94 (0.0542) | 5.6 (0.096) | 5.09 (0.0561) | 5.05 (0.0427) | 5.0 (0.0277) | 5.0 (0.0392) | 4.92 (0.0532) | 4.83 (0.0736) | 4.54 (0.0977) |
| Mistral-7B | 5.48 (0.0979) | 5.2 (0.0877) | 5.01 (0.0195) | 4.97 (0.0334) | 4.68 (0.0914) | 4.52 (0.0979) | 4.29 (0.0889) | 4.02 (0.0274) | 4.0 (0) | 4.0 (0) |
| DeepSeek-V3 | **10.35 (0.1695)** | **10.51 (0.1628)** | **10.76 (0.1841)** | **10.92 (0.1954)** | **11.21 (0.1551)** | **11.53 (0.1504)** | **11.67 (0.1594)** | **11.8 (0.1518)** | **11.88 (0.1337)** | **11.66 (0.1364)** |
| Random | 8.02 (0.1239) | 7.06 (0.0465) | 6.64 (0.1127) | 6.92 (0.1198) | 7.06 (0.1435) | 7.53 (0.1223) | 7.69 (0.1405) | 7.49 (0.1828) | 7.56 (0.1475) | 7.59 (0.1361) |

## D.3. Half-Width of Confidence Interval

In Figure 6, Table 9, Figure 7 and Table 10, we present the detailed experiment results for the half-width of the synthetic confidence interval $\mathcal{S}^{\mathsf{syn}}(\widehat{k})$.

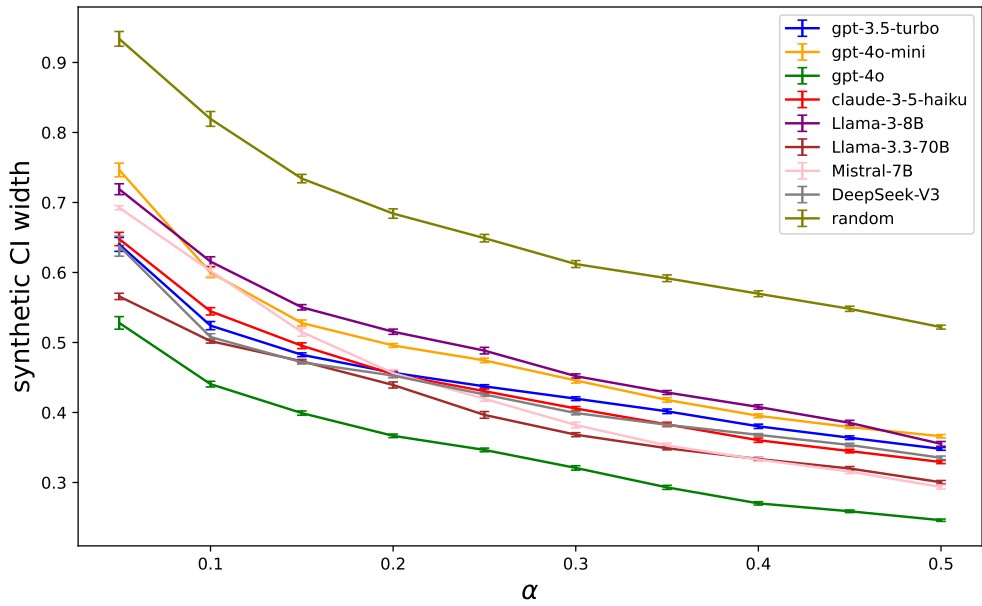

*Figure 6.* Confidence interval half-width for various LLMs and $\alpha$ over the OpinionQA dataset.

*Table 9.* Average half-width (with 95% margin of error in parentheses) of the synthetic confidence interval for various LLMs and $\alpha$ over the OpinionQA dataset. GPT-4o has the shortest confidence interval on average.

| $\alpha$ | 0.05 | 0.10 | 0.15 | 0.20 | 0.25 | 0.30 | 0.35 | 0.40 | 0.45 | 0.50 |
|---|---|---|---|---|---|---|---|---|---|---|
| GPT-3.5 -turbo | 0.6401 (0.0099) | 0.5241 (0.0058) | 0.4825 (0.0027) | 0.4568 (0.0021) | 0.4373 (0.0024) | 0.4198 (0.0027) | 0.4017 (0.0033) | 0.3803 (0.0031) | 0.364 (0.0026) | 0.348 (0.002) |
| GPT-4o -mini | 0.7465 (0.0097) | 0.5999 (0.0072) | 0.5279 (0.0043) | 0.4958 (0.0027) | 0.4745 (0.0032) | 0.4457 (0.0037) | 0.418 (0.0032) | 0.3953 (0.0028) | 0.3792 (0.0022) | 0.3661 (0.0024) |
| GPT-4o | **0.528 (0.009)** | **0.4405 (0.0041)** | **0.3991 (0.0031)** | **0.3667 (0.0025)** | **0.3466 (0.0026)** | **0.3208 (0.0031)** | **0.293 (0.0029)** | **0.2702 (0.0024)** | **0.2589 (0.0018)** | **0.2461 (0.0017)** |
| Claude-3-5 -Haiku | 0.6478 (0.0095) | 0.5446 (0.0052) | 0.4954 (0.0041) | 0.4548 (0.0035) | 0.4305 (0.0031) | 0.4057 (0.0028) | 0.383 (0.0031) | 0.3604 (0.0031) | 0.3449 (0.0026) | 0.3292 (0.0023) |
| Llama-3-8B | 0.7189 (0.0078) | 0.6156 (0.0069) | 0.5503 (0.0038) | 0.5154 (0.0038) | 0.4884 (0.0047) | 0.4519 (0.0032) | 0.4286 (0.0028) | 0.4079 (0.0032) | 0.3852 (0.0036) | 0.3552 (0.0034) |
| Llama-3.3 -70B | 0.5658 (0.0045) | 0.502 (0.0029) | 0.4732 (0.0026) | 0.4393 (0.0044) | 0.3965 (0.0048) | 0.3683 (0.0029) | 0.3489 (0.0023) | 0.3336 (0.0022) | 0.32 (0.0028) | 0.3004 (0.0025) |
| Mistral-7B | 0.6925 (0.0028) | 0.6024 (0.0079) | 0.5149 (0.0058) | 0.4563 (0.0047) | 0.4196 (0.0035) | 0.3821 (0.0041) | 0.3529 (0.0033) | 0.3328 (0.0026) | 0.3156 (0.0029) | 0.2935 (0.0028) |
| DeepSeek-V3 | 0.6376 (0.0143) | 0.5078 (0.0047) | 0.4717 (0.0025) | 0.4527 (0.0031) | 0.4257 (0.0026) | 0.3992 (0.0026) | 0.3824 (0.0019) | 0.3681 (0.0018) | 0.3536 (0.0025) | 0.3353 (0.0024) |
| Random | 0.9337 (0.0105) | 0.8194 (0.0106) | 0.7341 (0.006) | 0.6842 (0.0067) | 0.649 (0.0054) | 0.612 (0.005) | 0.5918 (0.0048) | 0.5697 (0.0042) | 0.5481 (0.0037) | 0.522 (0.0028) |

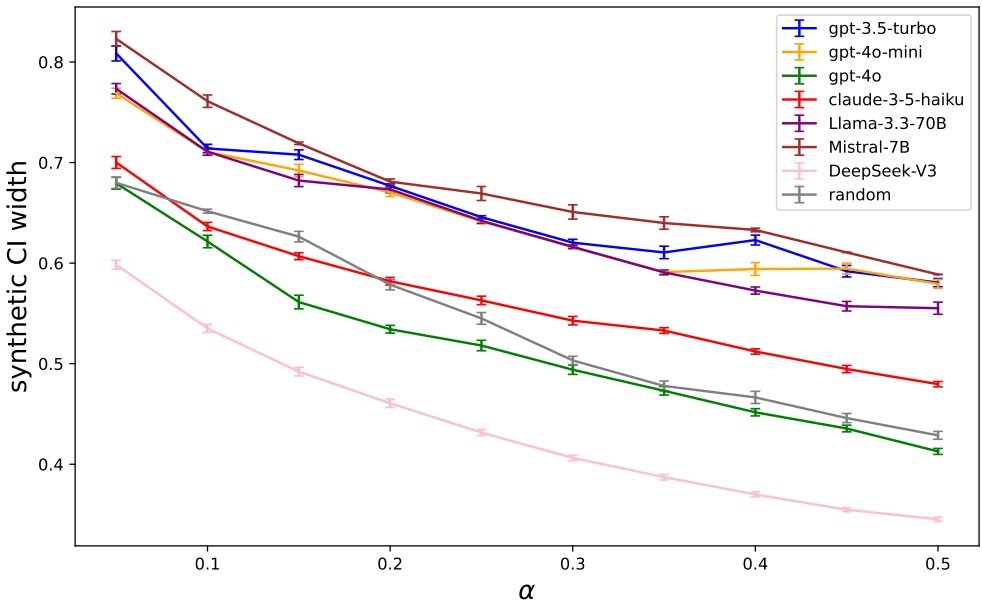

*Figure 7.* Confidence interval half-width for various LLMs and $\alpha$ over the EEDI dataset.

*Table 10.* Average half-width (with 95% margin of error in parentheses) of the synthetic confidence interval for various LLMs and $\alpha$ over the EEDI dataset. DeepSeek-V3 has the shortest confidence interval on average.

| $\alpha$ | 0.05 | 0.10 | 0.15 | 0.20 | 0.25 | 0.30 | 0.35 | 0.40 | 0.45 | 0.50 |
|---|---|---|---|---|---|---|---|---|---|---|
| GPT-3.5 -turbo | 0.8085 (0.0075) | 0.714 (0.0041) | 0.7078 (0.0048) | 0.6768 (0.002) | 0.6457 (0.0015) | 0.6203 (0.0034) | 0.6106 (0.0062) | 0.6229 (0.0049) | 0.592 (0.0057) | 0.5806 (0.0041) |
| GPT-4o -mini | 0.769 (0.005) | 0.7107 (0.0031) | 0.6922 (0.0061) | 0.6703 (0.004) | 0.6415 (0.0026) | 0.616 (0) | 0.5911 (0.0014) | 0.5941 (0.0064) | 0.5945 (0.0055) | 0.5794 (0.0043) |
| GPT-4o | 0.6794 (0.0059) | 0.6215 (0.0062) | 0.5613 (0.0068) | 0.5343 (0.0039) | 0.5181 (0.0052) | 0.494 (0.0047) | 0.4731 (0.0043) | 0.4517 (0.0036) | 0.4355 (0.0033) | 0.4128 (0.003) |
| Claude-3-5 -Haiku | 0.7001 (0.0059) | 0.6364 (0.0041) | 0.6069 (0.0035) | 0.582 (0.0038) | 0.5629 (0.0042) | 0.5428 (0.0042) | 0.533 (0.0028) | 0.5122 (0.0027) | 0.4947 (0.0036) | 0.4797 (0.0027) |
| Llama-3.3 -70B | 0.7734 (0.0052) | 0.7108 (0.0036) | 0.6821 (0.006) | 0.6733 (0.0033) | 0.6421 (0.0024) | 0.6162 (0.0018) | 0.5908 (0.0024) | 0.5727 (0.0036) | 0.5572 (0.0047) | 0.5551 (0.0061) |
| Mistral-7B | 0.823 (0.0073) | 0.761 (0.0062) | 0.7191 (0.0012) | 0.681 (0.0027) | 0.6693 (0.007) | 0.6509 (0.0071) | 0.6399 (0.0062) | 0.633 (0.0018) | 0.6107 (0) | 0.5887 (0) |
| DeepSeek-V3 | **0.5984 (0.0045)** | **0.5351 (0.004)** | **0.4921 (0.0043)** | **0.4607 (0.0042)** | **0.4315 (0.0031)** | **0.4063 (0.0027)** | **0.3872 (0.0029)** | **0.3699 (0.0025)** | **0.3548 (0.002)** | **0.3453 (0.0021)** |
| Random | 0.6799 (0.0056) | 0.6517 (0.002) | 0.6263 (0.0053) | 0.5786 (0.0052) | 0.545 (0.0058) | 0.5032 (0.0041) | 0.4779 (0.0049) | 0.4665 (0.0061) | 0.4459 (0.0045) | 0.4287 (0.0039) |

