# OpenReview forum: "Uncertainty Quantification for LLM-Based Survey Simulations"
_ICML.cc/2025/Conference — ICML 2025 poster_

### Official Review · Reviewer_XMzS · 2025-03-07

**Overall Recommendation:** 4

**Summary:**

This study investigates the reliable use of simulated survey responses generated by large language models  from the perspective of uncertainty quantification. The proposed approach transforms synthetic data into confidence intervals for human response group parameters, addressing distributional shifts between simulated and real populations. A key innovation is determining the optimal number of simulated responses: too many lead to overly narrow confidence intervals with poor coverage, while too few result in overly loose estimates. To address this, the method adaptively selects the sample size to ensure effective coverage on average. It is broadly applicable to any LLM, regardless of fidelity, and any confidence interval construction process. Additionally, the selected sample size quantifies the discrepancy between the LLM and the target population. The study demonstrates the effectiveness of this approach through experiments on real datasets and LLM-generated responses.

**Claims And Evidence:**

Yes

**Essential References Not Discussed:**

N/A

**Experimental Designs Or Analyses:**

Yes

**Methods And Evaluation Criteria:**

Yes

**Other Comments Or Suggestions:**

1. I think the figure 1 does not provide much useful information and may not be suitable for inclusion in the main body of the paper.
2. Adding a conclusion section would help readers quickly grasp the key findings of the study.

**Other Strengths And Weaknesses:**

### Strengths
1. The paper addresses a highly valuable problem. Given the large number of simulation experiments based on large language models, ensuring their reliability is a fundamental issue.
2. The paper provides a detailed formal discussion, offering a strong reference for future research.
3. The study includes comprehensive experiments.

### Weaknesses
1. I extremly expected to see some discussions on the nature and characteristics of language models for simulation, leading to methodological design and theoretical analysis based on these aspects. However, the paper does not seem to be structured in this way; instead, it treats the model more as a black box.
2. The paper starts by directly positioning "the number of synthetic samples" as the core research direction, which feels somewhat unnatural. This claim assumes that the distribution simulated by the language model changes with the number of samples and that there exists an optimal quantity range that aligns better with the real-world distribution. This conclusion is non-trivial and requires further justification.

**Questions For Authors:**

N/A

**Relation To Broader Scientific Literature:**

N/A

**Theoretical Claims:**

Yes

---

> ### Author Rebuttal · Authors · 2025-03-31
>
> We thank the reviewer for the thoughtful review and the encouraging score. Below are our responses.
>
> **Weaknesses**
>
> > *1. I extremely expected to see some discussions on the nature and characteristics of language models for simulation, leading to methodological design and theoretical analysis based on these aspects. However, the paper does not seem to be structured in this way; instead, it treats the model more as a black box.*
>
> The reviewer is correct that our method treats the LLM as a black box. An advantage is that the method works with LLMs with arbitrary fidelities and architectures. This ensures its wide applicability, as many LLMs are not open-source.
>
> > *2. The paper starts by directly positioning "the number of synthetic samples" as the core research direction, which feels somewhat unnatural. This claim assumes that the distribution simulated by the language model changes with the number of samples and that there exists an optimal quantity range that aligns better with the real-world distribution. This conclusion is non-trivial and requires further justification.*
>
> Thanks for raising the confusion. To perform simulation, one must specify a sample size $k$. Thus, it is an important quantity to determine.
>
> We would like to clarify that the distribution of the simulated responses does not change with the simulation sample size $k$. Instead, the sample size $k$ involves a trade-off between the width and the coverage validity of the confidence interval. Simulating too few samples leads to an overly wide confidence interval, while simulating too many samples causes the confidence interval to concentrate around the population parameter of simulated responses rather than the human responses. Our method is designed to find a simulation sample size $\widehat{k}$ that balances this trade-off.
>
> **Other Comments Or Suggestions**
>
> > *1. I think the figure 1 does not provide much useful information and may not be suitable for inclusion in the main body of the paper.*
>
> Thanks for the suggestion. We will move the figure to the appendix.
>
> > *2. Adding a conclusion section would help readers quickly grasp the key findings of the study.*
>
> Thanks for the suggestion. In the first two sentences of Section 5, we have summarized the main goal and methodology of our study. We will add the following sentence to summarize the key findings: "Numerical experiments on real datasets verified the coverage guarantees of our approach, and revealed that LLMs exhibited higher fidelity in simulating opinions to social problems than in simulating student responses to mathematics questions."

---

### Official Review · Reviewer_6eYT · 2025-03-14

**Overall Recommendation:** 2

**Summary:**

This study proposed a method to convert LLM simulated results into confidence sets for population parameters of human responses. It provides an estimate for the optimal number of simulations with theoretical proofs.

**Claims And Evidence:**

Claims:

(1) A general mathematical formulation of uncertainty quantification for LLM-based survey simulations was provided.

(2) The proposed method converts simulated responses into confidence sets for population parameters of human responses. It adaptively selects simulation sample size. This method is applicable for any LLMs, irrespective of its fidelity, and any methods for confidence set construction.

The first claim is true. I am not sure about the second claim as I do not fully understand the evaluation method. Additionally, the current study does not provide results for other confidence set construction.

**Essential References Not Discussed:**

References are satisfactory.

**Experimental Designs Or Analyses:**

Yes. However, I think this lacks a direct comparison of the simulated responses with \hat{k} and the real human responses.

**Methods And Evaluation Criteria:**

I am not fully following the evaluation of the selected sample size. Figures 2 and 3 show the average \hat{k} against different alphas. I understand that the number \hat{k} is the measure that the proposed study aims to predict. I do not understand why simply showing this value across different alphas demonstrates the validity of the proposed method.

Moreover, since this study aims to estimate the optimal number of simulations that can best represent the human response distributions, why not evaluate the method by actually simulating the responses \hat{k} times on the two datasets and compute the distribution differences between the simulated responses and real human responses?

**Other Comments Or Suggestions:**

I did not find typos.

**Other Strengths And Weaknesses:**

Strengths

(1) The research question is important. The potential contribution could be impactful as it may provide a principle in using LLMs for survey response simulation.

(2) This study provides theoretical support for selecting the number of simulations.

Weaknesses

(1) The main weakness lies in the clarity of the evaluation and lack of a more direct evaluation method. As aforementioned, it would be better if the authors could further elaborate on the evaluation of the selected simulation size. Additionally, it would strengthen the study if the simulated results using \hat{k} can be compared directly with the real human responses.

(2) It is not intuitively straightforward why the domains of the survey may not be a big factor when designing the approach. It would be better if this can be further discussed.

**Questions For Authors:**

Please see my main questions (concerns) for the evaluation method.

Other questions:

(1) Would different hyperparameter settings of the LLMs influence the final results (e.g., decoding strategy)?

**Relation To Broader Scientific Literature:**

The potential contribution would be impactful as this method would provide an effective way to select the number of simulations, saving the inference cost.

**Theoretical Claims:**

Yes, I checked the proof. Based on the assumptions, it seems correct. However, I have doubts in the assumption. I wonder if the domain of the surveys would influence the simulation results.

---

> ### Author Rebuttal · Authors · 2025-03-31
>
> We thank the reviewer for the thoughtful review. Below are our responses. We hope they address your questions and concerns.
>
> **Claims And Evidence**
>
> > *the current study does not provide results for other confidence set construction.*
>
> We clarify that the coverage guarantee (Theorem 3.3) of our general method in Section 3 holds for any confidence set construction procedures, such as inverting hypothesis tests, the bootstrap, and the empirical likelihood ratio function. In our experiments, we constructed confidence intervals based on Hoeffding's concentration inequality, because it is arguably the simplest construction procedure that has a valid coverage guarantee for any finite sample size.
>
> **Methods And Evaluation Criteria**
> > *I do not understand why simply showing this value [$\hat{k}$] across different alphas demonstrates the validity of the proposed method.*
>
> The reviewer is correct that the value of $\hat{k}$ does not demonstrate the validity of our method. Instead, we should verify the coverage guarantee of the induced confidence sets as claimed in our theorems. We examine this through the miscoverage proxy $\tilde{L}(\hat{k})$ in Equation (18). According to the explanations there, $\mathbb{E}[\tilde{L}(\hat{k})]\le\alpha$ implies a controlled miscoverage probability $\mathbb{P}(\theta(\psi)\not\in\mathcal{S}^{\mathsf{syn}}(\hat{k}))\le\alpha$. Thus, to check the validity of our method, we will perform a statistical hypothesis test on $H_0:\mathbb{E}[\tilde{L}(\hat{k})]\le\alpha$ against the alternative $H_1:\mathbb{E}[\tilde{L}(\hat{k})]>\alpha$. Due to space constraint, we kindly ask the reviewer to refer to the tables of $p$-values in our response to reviewer xDnM. In short, all $p$-values are reasonably large, supporting the null hypothesis $\mathbb{E}[\tilde{L}(\hat{k})]\le\alpha$ at, say, the 0.05 significance level. This shows that the miscoverage probability is effectively controlled by our method. We will add these results and discussions to the paper.
>
> >*... since this study aims to estimate the optimal number of simulations that can best represent the human response distributions, why not evaluate the method by actually simulating the responses $\hat{k}$ times on the two datasets and compute the distribution differences between the simulated responses and real human responses?*
>
> We thank the reviewer for raising this potential connection. Our method aims to select a simulation sample size $\hat{k}$ such that the confidence set $\mathcal{S}^{\text{syn}}(\hat{k})$ constructed from $\hat{k}$ simulated samples can cover the human population parameter with a prescribed probability while being as tight as possible. Therefore, a natural way to evaluate our method is to examine its coverage probability as well as the size of the confidence set. The results are summarized in Appendix C.4.2 and Table 3 in the original manuscript. Please also refer to our response to your comment in "Methods And Evaluation Criteria". Those results indeed reflect the distributional difference between the simulated and real responses: for instance, high coverage probability plus small size would certify small difference. We did not directly simulate $\hat{k}$ samples and directly compute the distributional differences, as it would require defining (additional) discrepancy metrics.
>
> **Weaknesses**
>
> > *(1) The main weakness lies in the clarity of the evaluation and lack of a more direct evaluation method ...*
>
> Please refer to our responses above.
>
> > *(2) It is not intuitively straightforward why the domains of the survey may not be a big factor when designing the approach.*
>
> It is true that our procedure for selecting the simulation sample size $\hat{k}$ as well as the validity of our approach does not rely on the specific survey domain. However, the survey domain does implicitly play a role in the outcomes of our approach. Different domains may have different target populations and formats of survey questions (e.g., the survey responses could be multiple-choice, numerical, or in a Likert scale). Moreover, the simulation power of the LLM can also be very different across different domains. All these factors can cause our method to choose different simulation sample sizes $\hat{k}$ and confidence sets $\mathcal{S}^{\text{syn}}(\hat{k})$, but the coverage guarantee will always hold.
>
> **Questions**
>
> > *(1) Would different hyperparameter settings of the LLMs influence the final results...?*
>
> The coverage guarantee of our method always holds regardless of the choice of LLM. On the other hand, the hyperparameter setting of the LLM does affect the quality of the LLM simulations. This will in turn affect the chosen simulation sample size $\hat{k}$ and the size of confidence set $\mathcal{S}^{\text{syn}}(\hat{k})$. For example, if a certain hyperparameter setting improves the quality of the LLM simulations, then we expect our method to select a larger simulation sample size $\hat{k}$, which gives a tighter confidence set.

---

### Official Review · Reviewer_RRnK · 2025-03-14

**Overall Recommendation:** 2

**Summary:**

The paper proposes a novel framework for converting synthetic survey responses generated by large language models (LLMs) into statistically valid confidence intervals for population parameters. By focusing on uncertainty quantification, the authors develop a principled method to adaptively select the simulation sample size (denoted as k̂) such that the confidence intervals maintain a prescribed coverage probability. The paper supports its contributions through rigorous theoretical guarantees—including coverage and sharpness results—and extensive experiments on two real-world datasets (EEDI and OpinionQA) using several state-of-the-art LLMs.

**Claims And Evidence:**

The paper makes several claims:

- It introduces a data-driven method to determine an effective sample size that produces confidence intervals with prescribed coverage probabilities.
- The method provides theoretical guarantees on both coverage and sharpness, implying that the selected sample size yields neither overly conservative nor excessively loose intervals.

While the theoretical analysis seems sound, the discussions focus on multiple-choice survey responses (with binary or limited choice scenarios), which raises questions about its generality. Given that LLMs excel at generating diverse and open-ended responses, this narrow focus does not fully leverage their potential in simulating human opinions.

**Essential References Not Discussed:**

Not that I know of.

**Experimental Designs Or Analyses:**

The experimental designs for the EEDI and OpinionQA datasets are methodologically sound within the limited scope of multiple-choice surveys.

It also relies on some assumptions such as the independence of survey questions and the applicability of CLT approximations, which would be beneficial if the authors could investigate the effect if those assumptions do not hold. Especially the independence of survey questions.

**Methods And Evaluation Criteria:**

- The scope of the paper is not clearly stated in the introduction or problem statement. The paper seems to focus only on multiple-choice survey, which is quite limited given the advantage of LLMs in human simulation is to produce opinions on open-ended questions. Therefore, the taste of LLMs in this paper is not very clear. The proposed method to estimate the sample size is also not tailored for LLMs.

- The method relies entirely on the base LLM to produce synthetic responses. It is well-known that LLMs can introduce significant bias—especially against minority groups—which is not adequately addressed in the current framework.

- A considerable portion of the theoretical and experimental work is devoted to the binary response setting, which may limit the practical applicability of the method to more complex survey scenarios.

**Other Comments Or Suggestions:**

The paper writing is difficult to follow in general. It would be better if the authors could clarify the scope of the paper earlier (in the introduction, for example).

**Other Strengths And Weaknesses:**

My main concerns are the limited scope of the paper with the main emphasis on determining the number of samples needed, neglecting the bias on underlying LLMs.

**Questions For Authors:**

Since we’re having access to human responses, can we use it to calibrate LLM responses to reduce bias?

**Relation To Broader Scientific Literature:**

The paper focuses on the problem of applying LLM for human simulation. However, the scope is limited since they only focus on determining the number of sample sizes needed, which has limited scope.

**Theoretical Claims:**

I haven't verified proofs in the paper.

---

> ### Author Rebuttal · Authors · 2025-03-31
>
> We thank the reviewer for the thoughtful review. Below are our responses. We hope they address your questions and concerns.
>
> **Claims And Evidence**
>
> >*... the discussions focus on multiple-choice survey responses... which raises questions about its generality.*
>
> Our motivating example in Section 2 focuses on binary responses for the purpose of illustration and ease of understanding. However, our general method and theory in Section 3 apply to a much broader range of survey questions. Specifically, they are applicable as long as the raw responses can be converted into a quantifiable metric. This includes the possibilities of text responses as well as a survey consisting of multiple related questions. Moreover, most existing works on assessing the fidelity of LLM simulations (e.g., those cited in "Related Works") focus on multiple-choice questions.
>
> **Methods And Evaluation Criteria**
>
> >*The proposed method ... is also not tailored for LLMs*
>
> The reviewer is correct that our method treats the LLM as a black box. An advantage is that the method works with LLMs with arbitrary fidelities and architectures. This ensures its wide applicability, as many LLMs are not open-source. In addition, our framework also applies to any black-box generative models beyond LLMs, such as image or video generators.
>
> >*LLMs can introduce significant bias...*
>
> We agree with the reviewer that LLMs can introduce significant bias against certain populations. However, as long as the target population is specified and we have human responses from that population, our method will construct a reliable confidence set for that population regardless of any bias the LLM may have. The intuition is that by comparing synthetic responses with human responses, we choose a suitable effective sample size $\hat{k}$ so that the bias of LLM is dominated by the random error associated with $\hat{k}$ synthetic samples. Therefore, the confidence set does take care of the LLM's bias for the given population.
>
> In addition, recent works have investigated mitigating LLMs' bias towards certain populations, through fine-tuning, prompting, etc. One may use these techniques to simulate more truthful responses from the LLM, and apply our method over these simulations to obtain tighter and more informative confidence sets.
>
> >*A considerable portion of the theoretical and experimental work is devoted to the binary response setting...*
>
> Please refer to our response to your comment in "Claims And Evidence".
>
> **Experimental Designs Or Analyses**
>
> >*It also relies on some assumptions such as the independence of survey questions and the applicability of CLT approximations...*
>
> In our experiments, there are at least 100 human responses per question. As a standard practice, we expect the CLT-based confidence intervals $\mathcal{S}_j$ to be accurate.
>
> The independence of survey questions is used for theoretical analysis only. It is similar to the assumption of i.i.d. data points in the statistical machine learning literature. We totally agree that it may not hold in practice, including the real-world datasets in our paper. However, our experiments show that even when this assumption may be violated, our method still achieves good coverage empirically.
>
> **Weaknesses**
>
> >*My main concerns are the limited scope of the paper... neglecting the bias on underlying LLMs.*
>
> Please refer to our response to your second comment in "Methods And Evaluation Criteria".
>
> **Other Comments Or Suggestions**
>
> >*The paper writing is difficult to follow in general. It would be better if the authors could clarify the scope of the paper earlier (in the introduction, for example).*
>
> Thanks for the suggestion. To clarify the main goal of our paper, we will change the first sentence of the abstract to "We investigate the use of large language models (LLMs) to simulate human responses to survey questions, and perform uncertainty quantification to gain reliable insights."
>
> To clarify the survey simulation procedure that our paper is concerned with, we will add the following sentence to the first paragraph of Section 1: "The typical simulation procedure consists in prompting an LLM with a real or fictional persona as well as a survey question, and collecting the LLM's responses."
>
> **Questions**
>
> >*Since we're having access to human responses, can we use it to calibrate LLM responses to reduce bias?*
>
> We believe that this is an important consideration in LLM simulation. In Section 5, we have mentioned the possibility of combing our approach with debiasing methods. As an example, one may first use part of the human data for LLM alignment, and then apply our method with the rest of the human data. However, this is beyond the scope of our current work, as our goal is to convert any black-box LLM simulations to reliable confidence intervals, regardless of the bias of the LLM. Thus, we will leave it as future work to use human data to debias LLM simulations.

---

> > ### Comment · Reviewer_RRnK · 2025-04-02
> >
> > We thank the authors for their detailed responses. However, my main concerns regarding the scope of the paper still remain. The proposed method is not specifically designed for LLMs, so the discussion focused on LLMs for the proposed method is not really necessary. The paper also neglects the inherent bias from LLM simulations, which I believe is a crucial problem for LLM simulations to be adopted. Therefore, I would like to keep my initial score.

---

> > > ### Author Response · Authors · 2025-04-03
> > >
> > > We thank the reviewer for the prompt reply. Below are our responses. We sincerely hope they address your concerns.
> > >
> > > > *The proposed method is not specifically designed for LLMs, so the discussion focused on LLMs for the proposed method is not really necessary.*
> > >
> > > We agree that our method applies to any black-box generative models beyond LLMs. This wide applicability of our approach is an advantage, given that many generative models are not open-sourced. We focus on LLMs due to their vital role in simulating human responses in many fields. Such practical importance makes them a natural motivating example and testbed for our method.
> > >
> > > > *The paper also neglects the inherent bias from LLM simulations, which I believe is a crucial problem for LLM simulations to be adopted.*
> > >
> > > As we pointed out in our previous rebuttal, our method is designed to handle *any bias* in the LLM simulations. Specifically, it can convert any imperfect and biased LLM simulations into reliable confidence intervals for population parameters of the human responses. The width of the confidence interval is adaptively chosen to dominate the bias, so that the coverage guarantees always hold. Our main results, Theorems 2.1 and 3.1, do not make any assumption on the bias and thus allow for arbitrary bias. Therefore, we respectfully disagree with the reviewer's claim that we neglect the LLM's bias.

---

### Official Review · Reviewer_xDnM · 2025-03-16

**Overall Recommendation:** 4

**Summary:**

**Survey Simulation Problem**: This paper proposes using LLMs to estimate the statistics for each survey question without asking humans to fill out the survey.  For example, in an educational test scenario, if we aim to estimate the probability $\mu$ of students correctly answering a test question, we can employ an LLM to sample multiple answers, and then we can use these answers to construct a confidence interval to estimate this probability.

**Methodology**:
- 1. Given a range of user profiles (e.g., gender, age), sample K user profiles.
- 2. For each survey question, combine it with K=50 user profiles, and prompt LLM to get K corresponding answers.
- 3. Calculate the confidence interval (the statistics we want) based on these K answers (e.g., Apply Hoeffding’s concentration inequality).
- 4. Utilize survey questions in the training dataset, and their corresponding real human answers to identify the optimal hyperparameter \( K \).
- 5. For a new survey question, follow steps (1)-(3) to compute the respective confidence interval.

## update after rebuttal
In general, I think the problem is important and practically useful, and the solution of applying conformal-prediction-style calibration to obtain a better-calibrated statistics seems quite fit in this case. Given these points, I am leaning towards acceptance.

**Claims And Evidence:**

See the following "Methods And Evaluation Criteria" and "Theoretical Claims" for details.
In general, the claims in this paper are supported by convincing evidence.

**Essential References Not Discussed:**

To the best of my knowledge, it appears to include the necessary references.

**Experimental Designs Or Analyses:**

Overall, the experimental design (train/test approach, empirical miscoverage checks) is reasonable and aligns with the proposed calibration method.
The issue is that most of the experimental results are put in the Appendix and some details are not shown, which quite disrupts the flow of the reasoning and understanding.

**Methods And Evaluation Criteria:**

**Method**

This paper's goal is to estimate a statistic for a given survey question—one that would require data from the real population—by instead simulating responses from an LLM. The challenge is that if we sample extensively from an LLM, the resulting estimate reflects the LLM’s internal distribution rather than the true population’s.

This paper handles this misalignment by using actual responses to similar questions for calibration. Then it compares each LLM-based confidence set against these real outcomes. If the LLM-based confidence sets repeatedly fail to include the real responses, which indicates a sizable mismatch between the synthetic and real distributions, then the algorithm will stop increasing the synthetic sample size (and hence interval narrowing) early, maintaining adequate coverage. In this way, it ensures, on average, the final confidence interval achieves near $1-\alpha$ coverage of the real statistic.

A limitation is that the coverage guarantees hold on average over random draws of survey questions and data, not necessarily for any particular question. In many practical applications, one does care about accuracy for each specific question. However, in the calibration literature, per-question calibration (i.e., sample-wise) is typically infeasible.

Overall, the proposed calibration procedure is still effective with a theoretical guarantee.

**Evaluation Criteria**
- The 2 datasets are representative: 1) use EEDI Dataset to estimate the probability a student answers a math question correctly and produce a confidence interval around that probability; 2) use OpinionQA to estimate the mean sentiment score for a given question and produce a confidence interval.
-  The evaluation metrics test miscoverage rate is reasonable.

**Other Comments Or Suggestions:**

See the section "Other Strengths And Weaknesses" above.

**Other Strengths And Weaknesses:**

**Strengths**
The paper addresses a relatively new and interesting task—survey response simulation—where large language models are used to estimate the real population's responses for a given questionnaire. More importantly, it provides a framework for estimating the survey’s statistical results using a confidence interval, which gives a quantifiable range with a $1-\alpha$ coverage guarantee.

**Weaknesses**
1. **Writing and Structure**: The paper’s presentation could be more accessible. Specifically, the abstract is difficult to parse initially; including a brief explanation or definition of the survey simulation task at the beginning would help readers understand the purpose immediately.
2. **Experimental Analyses**: While the test miscoverage rate is quite important for practical validation, it is relegated to the appendix. Similarly, the discussion about selecting the simulation sample size \(k\) (which indicates misalignment between the LLM and real data) might benefit from its own dedicated section or case study, rather than being nested within the "Experiment Results" section.

**Questions For Authors:**

1. When creating \(K\) user profiles based on a given range of features (e.g., gender, age), how exactly are these profiles sampled? From what I understand, it seems they are randomly drawn from the dataset where such profile information already exists. But if we have a survey question without corresponding real-data profiles, do you then sample feature values uniformly from the space of possible profiles? Or use some other method?

**Relation To Broader Scientific Literature:**

The authors’ work falls into the category of using LLMs as “synthetic subjects” in survey or experimental contexts. While several of those prior studies demonstrate that LLMs can be cost-effective in reproducing or approximating human‐subject responses, most do not provide formal guarantees on how well those simulations match real distributions. This paper’s core contribution lies in formalizing and ensuring coverage—i.e., constructing confidence sets around population estimates that remain valid despite misalignment between real and simulated data.

The authors extend ideas from uncertainty quantification (notably in distribution‐free or conformal settings) by adapting them to an LLM scenario, where the synthetic distribution is not guaranteed to reflect the true data‐generating process. Their approach incorporates “calibration data” (a set of questions with real responses) to select the simulation sample size, addressing a gap in earlier LLM-based studies that did not systematically gauge or correct for misalignment. In that sense, their approach parallels conformal inference’s reliance on real data for coverage calibration—except they repurpose it for a new “simulation from an imperfect distribution” challenge.

**Theoretical Claims:**

I briefly reviewed Theorems 2.6 and 3.3; they seem correct, though I might have overlooked some details.

---

> ### Author Rebuttal · Authors · 2025-03-31
>
> We thank the reviewer for the thoughtful review and the encouraging score. Below are our responses.
>
> **Experimental Designs Or Analyses**
> > *... most of the experimental results are put in the Appendix and some details are not shown...*
>
> Thanks for raising this point. We will add more experiment details and results in the main text, especially those verifying our guarantees on the coverage probability. In particular, we will examine the miscoverage proxy $\tilde{L}(\hat{k})$ in Equation (18) of the paper. As we explain in the paper, $\mathbb{E}[\tilde{L}(\hat{k})]\le\alpha$ implies a controlled miscoverage probability $\mathbb{P}(\theta(\psi)\not\in\mathcal{S}^{\text{syn}}(\hat{k}))\le\alpha$. Thus, to check the validity of our method, we will perform a statistical hypothesis test on $H_0:\mathbb{E}[\tilde{L}(\hat{k})]\le\alpha$ against the alternative $H_1:\mathbb{E}[\tilde{L}(\hat{k})]>\alpha$. The following tables report the $p$-values computed from the one-sided $z$-test for different LLMs and $\alpha$ over the 100 random splits. We see that all $p$-values are reasonably large, supporting the null hypothesis $\mathbb{E}[\tilde{L}(\hat{k})]\le\alpha$ at, say, the 0.05 significance level. This shows that the miscoverage probability is effectively controlled by our method.
>
> OpinionQA:
>
> |min|25% quantile|median|75% quantile|max|
> |:---:|:---:|:---:|:---:|:---:|
> |0.03|0.25|0.51|0.70|0.99|
>
> |$\alpha$|0.05|0.10|0.15|0.20|0.25|0.30|0.35|0.40|0.45|0.50|
> |:---:|:---:|:---:|:---:|:---:|:---:|:---:|:---:|:---:|:---:| :---:|
> |GPT-3.5-turbo|0.06|0.13|0.20|0.22|0.70|0.35|0.34|0.26|0.78|0.84|
> |GPT-4o-mini|0.16|0.16|0.07|0.24|0.47|0.44|0.20|0.12|0.57|0.38|
> |GPT-4o|0.37|0.32|0.49|0.22|0.58|0.44|0.23|0.34|0.97|0.92|
> |Claude-3.5-Haiku|0.39|0.59|0.59|0.46|0.70|0.60|0.74|0.89|1.00|0.98|
> |Llama-3-8B|0.76|0.44|0.26|0.61|0.57|0.42|0.70|0.61|0.74|0.52|
> |Llama-3.3-70B|0.55|0.65|0.77|0.59|0.71|0.72|0.61|0.35|0.70|0.53|
> |Mistral 7B|0.62|0.05|0.05|0.03|0.65|0.60|0.54|0.21|0.34|0.34|
> |DeepSeek-V3|0.07|0.10|0.08|0.15|0.51|0.23|0.17|0.26|0.35|0.30|
> |Random|0.78|0.62|0.75|0.66|0.96|0.99|0.93|0.80|0.93|0.98|
>
> Only one $p$-value (Mistral with $\alpha=0.2$) is below 0.05. This is not surprising due to the multiple comparison problem. Indeed, even when the null hypothesis is true, one would expect to see one out of 20 $p$-values to be below 0.05.
>
> EEDI:
> |min|25% quantile|median|75% quantile|max|
> |:---:|:---:|:---:|:---:|:---:|
> |0.18|1.00|1.00|1.00|1.00|
>
> |$\alpha$|0.05|0.10|0.15|0.20|0.25|0.30|0.35|0.40|0.45|0.50|
> |:---:|:---:|:---:|:---:|:---:|:---:|:---:|:---:|:---:|:---:|:---:|
> |GPT-3.5-turbo|1.00|1.00|1.00|1.00|1.00|1.00|1.00|1.00|1.00|1.00|
> |GPT-4o-mini|1.00|1.00|1.00|1.00|1.00|1.00|1.00|1.00|1.00|1.00|
> |GPT-4o|0.18|0.58|1.00|0.97|1.00|0.98|0.93|0.83|0.98|0.99|
> |Claude-3.5-Haiku|0.87|1.00|1.00|1.00|1.00|1.00|1.00|1.00|1.00|1.00|
> |Llama-3.3-70B|0.96|1.00|1.00|1.00|1.00|1.00|1.00|1.00|1.00|1.00|
> |Mistral 7B|1.00|1.00|1.00|1.00|1.00|1.00|1.00|1.00|1.00|1.00|
> |DeepSeek-V3|0.94|0.98|0.98|1.00|1.00|1.00|1.00|1.00|1.00|1.00|
> |Random|0.98|1.00|1.00|1.00|1.00|0.99|1.00|1.00|0.99|1.00|
>
> **Weaknesses**
> > *1. ...the abstract is difficult to parse initially; including a brief explanation or definition of the survey simulation task at the beginning would help ...*
>
> Thanks for the suggestion. We will change the first sentence of the abstract to "We investigate the use of large language models (LLMs) to simulate human responses to survey questions, and perform uncertainty quantification to gain reliable insights."
>
> > *2. While the test miscoverage rate is quite important for practical validation, it is relegated to the appendix. Similarly, the discussion about selecting the simulation sample size ... might benefit from its own dedicated section ...*
>
> Thanks for the suggestions. We will include results on the miscoverage rate in the main text. Please refer to our response to your comment in "Experimental Designs Or Analyses" for more details. We will also move the results on the chosen simulation sample size $\hat{k}$ to a separate part.
>
> **Questions**
>
> > *When creating (K) user profiles based on a given range of features (e.g., gender, age), how exactly are these profiles sampled? ...*
>
> The reviewer is correct that in our experiments, the profiles are randomly drawn from the dataset, where we have access to the profile information. When real-data profiles are not available, practitioners may rely on their knowledge of the target population to create fictional profiles. We point out our method does not require the profile distribution $\mathcal{P}^{\mathsf{syn}}$ used for simulation to be the same as the target population $\mathcal{P}$, and automatically addresses any distribution shift between $\mathcal{P}^{\mathsf{syn}}$ and $\mathcal{P}$. Our Theorems 2.4 and 3.3 on the coverage guarantees do not assume any relation between them. Of course, we expect the confidence sets to be tighter when $\mathcal{P}^{\mathsf{syn}}$ and $\mathcal{P}$ are close.

---

> > ### Comment · Reviewer_xDnM · 2025-04-07
> >
> > Thank you for the reply! In general, I think the problem is important and practically useful, and the solution of applying conformal-prediction-style calibration to obtain a better-calibrated statistics seems quite fit in this case. Given these points, I am leaning towards acceptance.
> >
> > Besides, I am curious what would be the possible solution/adjustment if the test dataset has an out-of-distribution shift from the training dataset, and the conformal prediction framework will not work elegantly.

---

> > > ### Author Response · Authors · 2025-04-07
> > >
> > > We thank the reviewer for the positive feedback! Below is our response.
> > >
> > > > *Besides, I am curious what would be the possible solution/adjustment if the test dataset has an out-of-distribution shift from the training dataset, and the conformal prediction framework will not work elegantly.*
> > >
> > > The distribution shift problem is indeed an important consideration. In practice, there can be changes in the distributions of the survey questions and the human responses to those questions. The conformal prediction literature has developed a number of approaches to deal with distribution shift, such as reweighing the data [1] and adjusting the miscoverage level [2]. We believe it is possible to adapt those approaches to our framework, and leave it as future work.
> > >
> > > **References**
> > >
> > > [1] Barber, R. F., Candes, E. J., Ramdas, A., & Tibshirani, R. J. (2023). Conformal prediction beyond exchangeability. *The Annals of Statistics, 51*(2), 816-845.
> > >
> > > [2] Gibbs, I., & Candes, E. (2021). Adaptive conformal inference under distribution shift. *Advances in Neural Information Processing Systems, 34*, 1660-1672.

---

### Decision · Program_Chairs · 2025-05-01

**Decision:**

Accept (poster)

**Comment:**

The reviewers were divided on this paper, so I have also carefully reviewed it. Having published multiple times in the area of silicon sampling, I consider myself well qualified to assess it, and I was quite happy with the paper and its contributions, so I recommend acceptance.

The paper is solid. It addresses an important problem; it suggests a clean and workable solution; it nicely blends theory and experiment; and is well-written.

My only hesitation is that the the calibration questions must bear some similarity to the target question; assessing that similarity in any sort of formal way is a hard problem. However, I believe the strengths of the paper outweigh that weakness.